# Differential dynamics of cortical neuron dendritic trees revealed by long-term in vivo imaging in neonates

Shingo Nakazawa[1,2], Hidenobu Mizuno[1,2,3] & Takuji Iwasato[1,2]

Proper neuronal circuit function relies on precise dendritic projection, which is established through activity-dependent refinement during early postnatal development. Here we revealed dynamics of dendritic refinement in the mammalian brain by conducting long-term imaging of the neonatal mouse barrel cortex. By "retrospective" analyses, we identified "prospective" barrel-edge spiny stellate (SS) neurons in early neonates, which had an apical dendrite and primitive basal dendrites (BDs). These neurons retracted the apical dendrite gradually and established strong BD orientation bias through continuous "dendritic tree" turnover. A greater chance of survival was given to BD trees emerged in the barrel-center side, where thalamocortical axons (TCAs) cluster. When the spatial bias of TCA inputs to SS neurons was lost, BD tree turnover was suppressed, and most BD trees became stable and elaborated mildly. Thus, barrel-edge SS neurons could establish the characteristic BD projection pattern through differential dynamics of dendritic trees induced by spatially biased inputs.

[1] Division of Neurogenetics, National Institute of Genetics (NIG), Mishima, Shizuoka 411-8540, Japan. [2] Department of Genetics, SOKENDAI (The Graduate University for Advanced Studies), Mishima, Shizuoka 411-8540, Japan. [3] International Research Center for Medical Sciences (IRCMS), Kumamoto University, Kumamoto, 860-0811, Japan. Correspondence and requests for materials should be addressed to T.I. (email: tiwasato@nig.ac.jp)

A central question in neuroscience is how cortical circuits are established during postnatal development, wherein initial primitive circuits are refined by periphery-derived neural activity[1–3]. Dendritic projection patterns of neurons define the types of information the neuron receives and processes; thus, it is critical to understand how specific dendritic projection patterns are established during development[4,5]. However, the dynamic nature of developmental dendritic refinement, in which cortical neurons extend their dendrites toward specific sets of presynaptic axons, has remained largely unexplored.

Two-photon microscopy has been widely applied for in vivo imaging of structural plasticity in adult neuronal circuits, such as spine formation/elimination associated with learning and memory[6,7]. For studies of structural plasticity in the developing mammalian brain, two-photon in vivo imaging has been used to monitor morphological dynamics of dendritic spines and filopodia of cortical pyramidal neurons[8,9], climbing fiber axons in the cerebellum[10], thalamocortical and Cajal–Retzius axons in cortical layer 1[11], and so on. In contrast, technical difficulties have hindered the use of in vivo time-lapse (TL) imaging in studies on the dynamics of dendritic development in mammals, which occurs during early postnatal development. Major difficulties include (1) sparse yet intense in vivo labeling of neurons, which is necessary for visualization of detailed dendritic morphology, (2) in vivo labeling of specific axons that are presynaptic to dendrites of an identified neuron, and (3) use and maintenance of fragile newborn mice during in vivo imaging sessions. For these reasons, instead of in vivo imaging, acute or chronic slice culture has been predominantly used for two-photon or confocal TL imaging of dendritic development (e.g., dendritic arborization) in the mammalian cortex[12], hippocampus[13,14], and cerebellum[15]. Alternatively, transparent vertebrates, such as Xenopus tadpoles[16,17] and zebrafish larvae[18], have been used for in vivo TL imaging of dendritic development in the brain and/or retina.

To analyze dendritic refinement of cortical neurons, we have used layer 4 (L4) spiny stellate (SS) neurons in the mouse primary somatosensory cortex (barrel cortex) as a model[19–22]. This area contains "barrels", which are morphological and functional modules arranged to correspond with facial whiskers[23] (Supplementary Fig. 1a). SS neurons located around the barrel edge (edge-SS or eSS neurons) extend their basal dendrites (BDs) asymmetrically toward the barrel center, where termini of thalamocortical axons (TCAs) transmitting information from a single whisker form a cluster[24,25]. These characteristic barrel morphologies are formed during the first postnatal week in a periphery-derived input-dependent manner[26–28], making them a key model of the developmental refinement of cortical circuits. In particular, BD refinement of eSS neurons is an excellent model because in the mouse barrel cortex, an eSS neuron has appropriate presynaptic TCAs only at the side of the barrel center. Therefore, it is possible to quantitatively analyze dendritic projection specificity as BD orientation bias toward the barrel center.

As a first step toward in vivo imaging of dendritic refinement in the mammalian brain, we recently developed two methods: (1) the "Supernova" system, which allows sparse yet bright in vivo labeling of cortical L4 neurons when used in combination with in utero electroporation-based gene transfection[29,30], and (2) TCA-GFP transgenic (Tg) mice, which allows in vivo labeling of TCAs[30]. These innovations have allowed us to conduct short-term (18-h) in vivo imaging of SS neuron BDs starting at postnatal day 5 (P5)[30]. This study provided the first in vivo observation of dendritic dynamics in the neonatal mammalian brain, in which small-scale dynamics (i.e., elongation/retraction of dendritic branches) were analyzed. However, this study was far from sufficient, because dendritic refinement was already nearly completed at P5[30]. Therefore, to fully understand dynamic

mechanisms of BD refinement of SS neurons, long-term (over days) imaging starting at earlier neonatal stages was awaited.

In the current study, we have succeeded in vivo visualization of SS neuron dendrites in mouse pups as early as P3. We also developed a system by which neonatal mice as early as P3 grow up with adequate maternal care during imaging sessions over days. Based on these preparations, we here achieved long-term (3-d-long) imaging of the SS neuron dendrites starting at P3 and characterized large-scale dendritic dynamics (e.g., emergence/elimination of "dendritic trees"). Our long-term imaging in neonatal mice revealed the selection dynamics by which only a fraction of inner BD trees become "winners", exhibiting high stability and extensive elaboration. Also, we revealed the role of spatial patterns of periphery-derived inputs in the selection dynamics of BD refinement.

In addition, it is also important that our long-term imaging of the same neurons over time allowed for "retrospective" identification of prospective SS neurons during early neonatal stages, such as P3. In these stages, most SS neurons in mammals show similar morphological features as another type of L4 excitatory neurons, star pyramid (SP)[31]. Our system can use information obtained at later developmental stages (e.g., P6) to identify prospective SS neurons at earlier stages, which enabled us for the first time to characterize features of SS neurons during early neonatal stages. Our retrospective analyses also revealed the presence of two phases in eSS neuron BD refinement during neonatal stages. Thus, our novel in vivo imaging system contributes to our understanding of developmental mechanisms of cortical maturation in neonates.

## Results

### Long-term in vivo imaging of neonatal L4 neurons over 3 days.
In L4 of the developing mouse barrel cortex, whisker-specific TCA clusters emerge in the barrel center around P3[30], and eSS neurons acquire their characteristic BDs, which are asymmetrically extended within single barrels, primarily by P6[30,32]. Therefore, in the present study, to characterize the detailed time course of L4 SS neuron BD refinement, we sought to perform long-term in vivo TL imaging starting at P3 or earlier and ending at P6 (Fig. 1a). To visualize the detailed dendritic morphology of individual L4 neurons, L4 neurons were sparsely labeled with RFP using our in utero electroporation-based Supernova method[29,30] (Supplementary Fig. 1c). To visualize the barrel map in vivo, TCA-GFP Tg mice expressing EGFP in TCAs[30] were used (Supplementary Fig. 1b).

To achieve long-term neonatal imaging over 3 d, it is necessary for pups to have maternal care between imaging sessions. Maternal care is important not only to give pups sufficient nutrition but also to supply them with natural whisker inputs[33] and reduce separation stress that could affect brain development[34]. However, in initial trials most pups (63%; 5/8 pups) were neglected, killed, and/or injured by mothers within a day after surgery. To solve this problem, we first designed an extremely small titanium bar to help minimize discomfort of mothers during breastfeeding (Fig. 1b). Second, we selected foster mothers that showed good results in nursing neonatal mice that had a cranial window/titanium bar. These improvements increased the probability of pups that underwent surgery at P3 (90%; 19/21 pups) or P2 (73%; 8/11 pups) were nursed normally until P6, at which time, we terminated observations (Fig. 1c).

We then performed imaging of L4 neurons repeatedly every 8 h from late P3 (P3$_L$; around 8 p.m.) to late P5 (P5$_L$) and at late P6 (P6$_L$) (Fig. 1a, d–i). In a few cases, we also started imaging at late P2 (P2$_L$). Because L4 neurons were sparsely labeled and the

relative positions of neurons were roughly conserved, it was easy to identify the same neurons in images taken at different time points (imaging sessions) (Fig. 1d, g). All neurons (70 neurons from 5 mice) observed at $P3_L$ were present at $P6_L$, indicating that there was no cell death during imaging sessions.

TL-imaged pups significantly increased in body weight from $P3_L$ to $P6_L$, as did control pups, and there was no significant difference between TL and control pups even at $P6_L$ (Fig. 1j). We also found the barrel field size of $P6_L$ TL pups was larger than that of $P3_L$ controls and similar to that of $P6_L$ controls (Fig. 1k), in which cortical tangential sections prepared from TL pups

immediately after the $P6_L$ imaging and normal pups at $P3_L$ and $P6_L$ were used. The total BD length and BD tip number of L4 neurons in TL pups increased 3.0-fold and 2.7-fold, respectively, between $P3_L$ and $P6_L$ (Fig. 1l and Supplementary Fig. 1e), and both were similar between TL and control pups at $P6_L$ (Fig. 1m and Supplementary Fig. 1f). These analyses were done by in vivo imaging of non-TL pups (control), to which a cranial window/titanium bar was attached at P6, and TL pups at $P6_L$. Taken together, these results demonstrate that long-term imaging over 3–4 days starting at P3 or P2 causes no obvious abnormalities in brain development through $P6_L$.

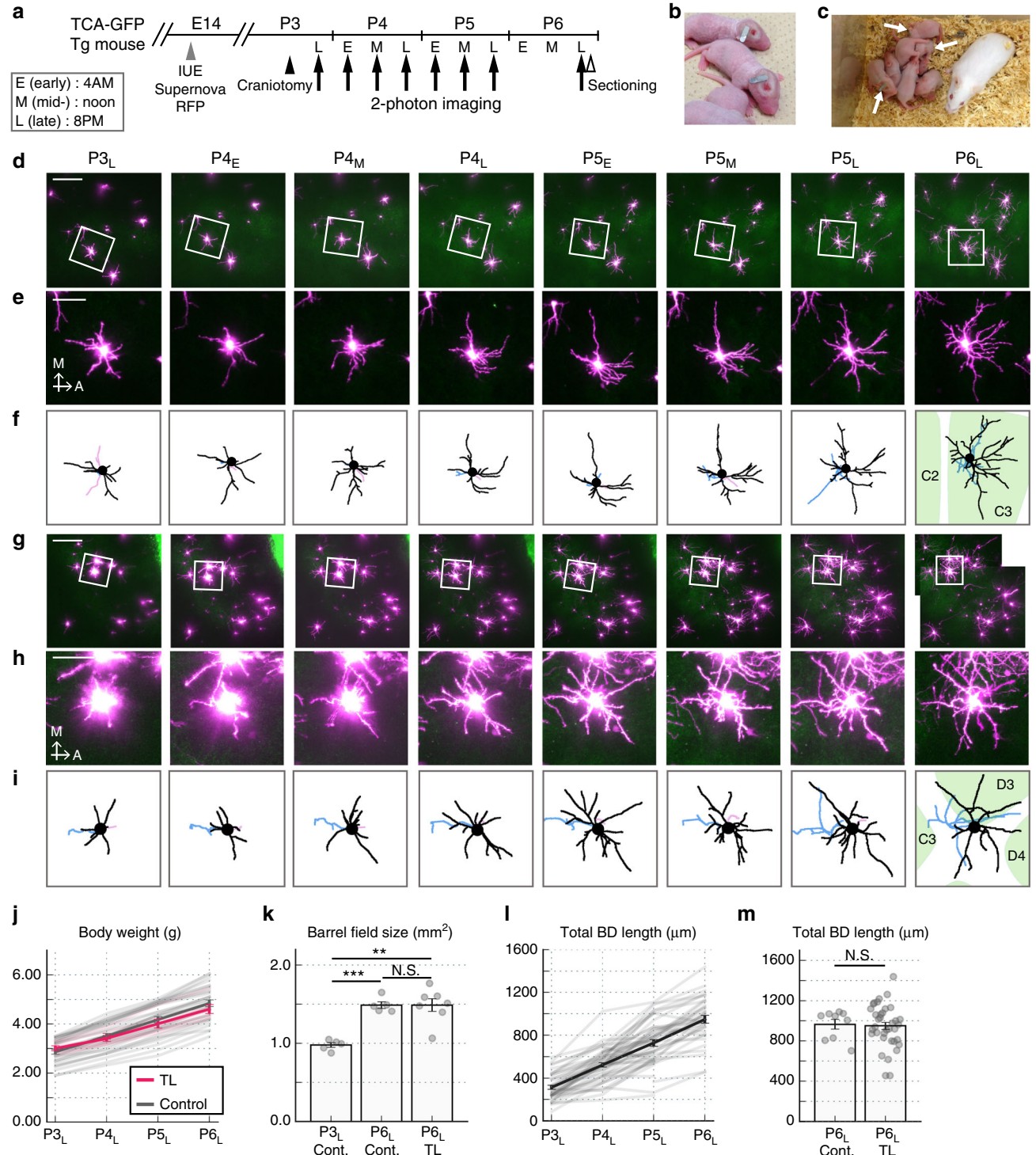

**Two types of neurons classified by apical dendrite (AD) dynamics**. When we compared dendritic morphologies of the same L4 neurons at $P3_L$ and $P6_L$, a striking difference was observed in AD (Fig. 2a). Most (97%, 36/37) L4 neurons had AD (29.03–228.29 μm long) at $P3_L$, and majority of them lost their AD during later development; at $P6_L$, 54% (20/37) had no AD. Analysis of daily changes in AD length by comparing images of the same L4 neurons taken at $P3_L$, $P4_L$, $P5_L$, and $P6_L$ (also at $P2_L$ in some cases) allowed us to classify individual L4 neurons into two groups (Groups 1 and 2). Neurons which shortened AD during imaging sessions were classified as Group 1 (Fig. 2b and Supplementary Fig. 2). In some neurons, AD was extended prior to initiation of retraction. We also found a neuron that had no AD at $P3_L$ (Supplementary Fig. 2e), which was also classified as Group 1. On the other hand, neurons that continuously extended AD throughout imaging sessions were classified as Group 2 (Fig. 2b and Supplementary Fig. 2). Based on these criteria, neurons shown in Fig. 1e and f and Fig. 1h and i were Group 1 and Group 2 neurons, respectively. Figure 2c and d also shows representative Group 1 and Group 2 neurons, respectively. Among 51 L4 neurons analyzed, 76% (39 neurons) and 24% (12 neurons) were categorized as Group 1 and Group 2 neurons, respectively (Fig. 2b and Supplementary Table 1).

We compared morphological aspects of BDs between Group 1 and Group 2 neurons. Increases in the total BD length were not different between the two groups (approximately 3-fold increase in both groups) (Fig. 2e). We then analyzed BD orientation bias toward the barrel center of these neurons using an orientation bias index (OBI) (see Supplementary Fig. 1d and Methods for details). For these analyses, we used only neurons located at the barrel edge (Fig. 2i and Supplementary Table 1). We found that at $P6_L$, the OBI of Group 1 neurons was quite high (Fig. 2j), indicating that their BDs showed strong orientation bias toward the barrel side, while the OBI of Group 2 neurons was close to 0.5 (Fig. 2j), indicating their BDs did not show orientation bias.

In the mature mouse barrel cortex, the majority (65–80%) of L4 neurons are SS neurons with no AD and multiple BDs projecting specifically within a single barrel. The other 20–35% are SP neurons with an AD and multiple BDs showing no orientation bias[24,35,36]. As described above, 76% of L4 neurons were Group 1 and 24% were Group 2. Group 1 neurons possessed no (or short) AD and their BDs exhibited strong orientation bias at $P6_L$, while Group 2 neurons possessed a long AD and their BDs exhibited no orientation bias. These characteristics of Group 1 and Group 2 neurons were consistent with those of SS and SP neurons, respectively. Therefore, hereafter, we refer to Group 1 neurons as SS neurons and Group 2 neurons as SP neurons, although we could not exclude the possibility that Group 2 may

contain a few SS neurons whose maturation was slower than others.

A recent histological study primarily using the ferret visual cortex demonstrated that virtually all L4 excitatory neurons in early postnatal stages have pyramidal shapes with a long AD and primitive BDs[31]. This was confirmed in the current study in the mouse barrel cortex. We also found that prospective SS neurons that exhibit a pyramidal shape lose their AD during development, providing direct evidence for developmental sculpting of SS neuron AD, as proposed by Callaway and Borrell[31]. In addition, we report three new findings. First, the AD was lost from SS neurons by gradual retraction (Fig. 2b, c). Second, the initiation timing and velocity of AD retraction varied among neurons even in the same animals (Supplementary Fig. 2). Third, once ADs started to retract, they did not extend again at least in our observation (Supplementary Fig. 2).

**Retrospective characterization of prospective eSS neurons**. Since SS neurons initially had an AD similar to that of SP neurons, it is possible that initiation of AD retraction in SS neurons may precede and/or trigger morphological differentiation. We examined this possibility by taking advantage of "retrospective" analyses. As described above, our long-term TL imaging allowed us to retrospectively distinguish SS and SP neurons even before the initiation of AD retraction (Fig. 2b), which appears impossible using conventional methods, such as histology.

To elucidate whether SS and SP neurons exhibit morphological differences in early neonatal stages, it is useful to compare the BD orientation bias of SS and SP neurons at $P3_L$. However, at this age, the TCA-GFP signal was too weak to clearly visualize the barrel boundary in vivo. In addition, relative positions of SS neurons could shift slightly between $P3_L$ and $P6_L$ due to brain enlargement (Fig. 1k) and possibly from SS neuron tangential migration. These issues hindered determination of the precise barrel boundary at $P3_L$. Therefore, in the current study, we used the simple version of OBI, which divides BD segment length in the barrel-center half (inside) by total BD length (see Supplementary Fig. 1d and Methods for details). The relative positions of individual L4 neurons were roughly conserved between $P3_L$ and $P6_L$ (e.g., Fig. 1d, g), therefore we could assume that the barrel-center direction from each L4 neuron was also conserved. Approximate barrel-center direction was determined by barrel map visualized at later developmental stages (e.g., $P6_L$). In this analysis, only neurons located at the barrel edge at $P6_L$ (eSS and barrel-edge SP (eSP) neurons) were included.

Intriguingly, we found the OBI of eSS neuron BDs was significantly larger than that of eSP neuron BDs at $P3_L$ (Fig. 2g), suggesting that eSS neurons already had BD orientation bias

**Fig. 1** Long-term in vivo imaging of cortical L4 neurons in neonates. **a** Schematic drawing of the in vivo time-lapse imaging between P3 and P6. IUE in utero electroporation. **b** A very light (~20 mg) and small (7 × 2 × 0.5 mm) titanium bar was used. **c** Representative image of pups (at P6) used for 3-d-long imaging (time-lapsed (TL) pups) (arrows). During imaging intervals, TL pups received maternal care with other littermates. **d** Representative Z-stack images of Supernova-RFP-labeled L4 neurons (magenta) and EGFP signals in TCA-GFP Tg mice. **e** Higher-magnification images of the neuron shown in (**d**) (square). A anterior, M medial. **f** Basal dendrites (BDs: black), apical dendrite (AD: red), and axon (blue) of the neuron were traced and reconstructed in three dimensions. Barrel map (green) determined by EGFP signal was shown (see Supplementary Fig. 1c). **g–i** Another set of representative case corresponding to (**d–f**). **j** Body weight change of TL ($n = 6$) and control ($n = 29$) pups (mean ± SEM (dense) and individual (faint)). $P3_L$-TL vs. $P6_L$-TL: $p < 0.001$, $g = 4.369$, Paired $t$-test. $P6_L$-TL vs. $P6_L$-control: $p = 0.312$, $g = 0.348$, Welch's $t$-test. **k** Barrel field areas were measured in tangential sections prepared from TL pups ($n = 7$) immediately after $P6_L$ imaging. As controls, normal pups at $P3_L$ ($n = 5$) and $P6_L$ ($n = 5$) were used. $P3_L$-control vs. $P6_L$-control: $p < 0.001$, $g = 6.131$; $P3_L$-control vs. $P6_L$-TL: $p = 0.001$, $g = 2.913$; $P6_L$-control vs. $P6_L$-TL: $p = 0.995$, $g = 0.003$; Welch's $t$-test with Holm's correction. **l** Increase of total BD length of TL pup L4 neurons during 3-d-long imaging ($n = 51$ neurons from eight mice). Mean ± SEM (dense) and values of individual neurons (faint) were shown. **m** Total BD length of L4 neurons of TL ($n = 36$ neurons from six mice) and control ($n = 9$ neurons from two mice)

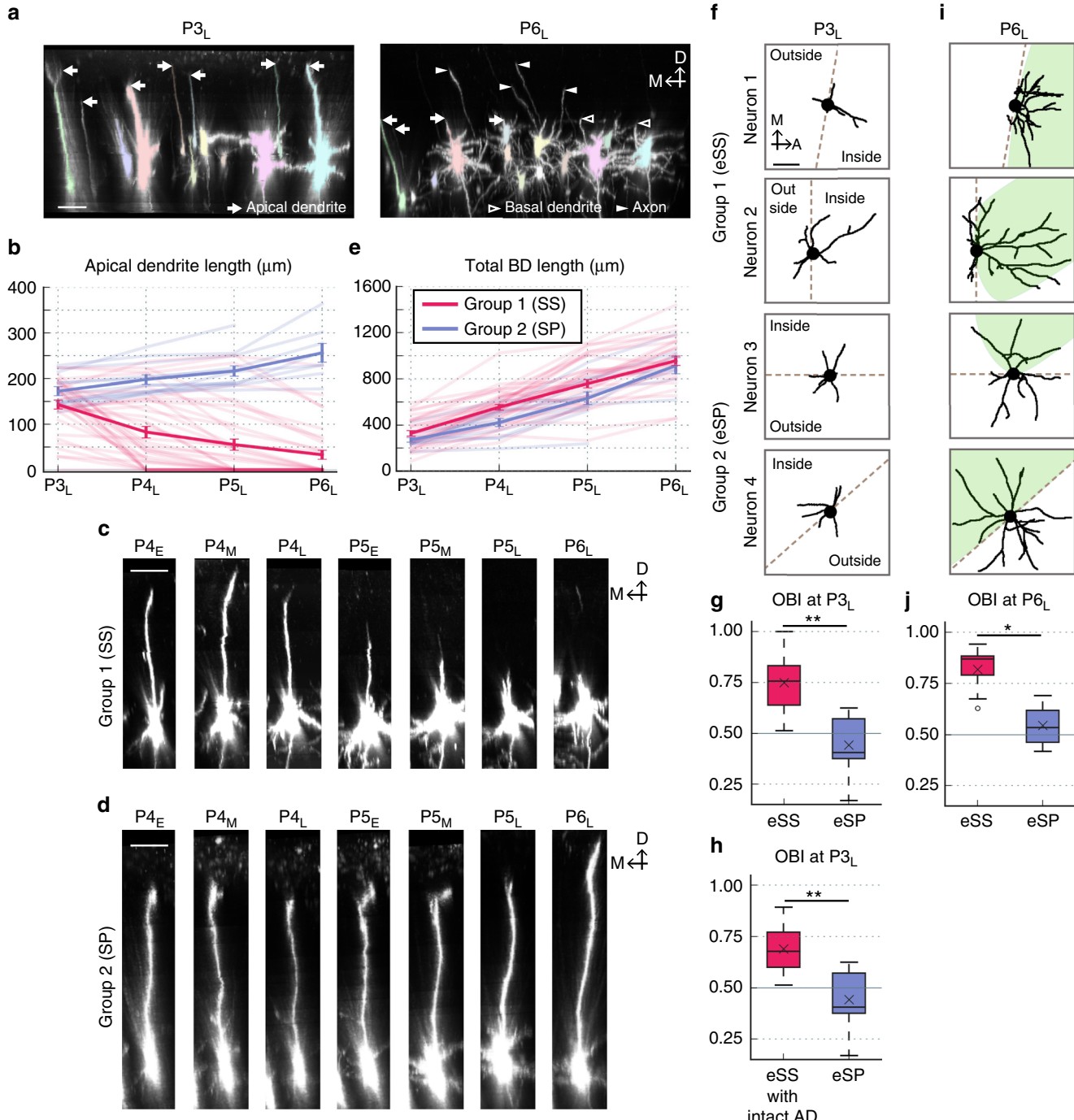

**Fig. 2** AD dynamics and morphological features of L4 neurons in early neonates. **a** Most L4 neurons possessed an AD at P3$_L$, while the majority of these neurons had shortened ADs by P6$_L$. The same neurons are colored the same. Arrows, AD tips; filled arrowheads, axons; open arrowheads, BD tips. M medial, D dorsal. **b** Plots of the mean ± SEM (dense) and individual (faint) of AD length of Group 1 (red) and Group 2 (blue) neurons (see also Supplementary Fig. 2). **c, d** Examples of AD dynamics in Group 1 (**c**) and Group 2 (**d**) neurons. D dorsal. **e** The mean ± SEM (dense) and individual (faint) plots of total BD length of Group 1 (red) and Group 2 (blue) neurons. Sample sizes for **b** and **e** are shown in the Methods. **f** Representative BD traces of two Group 1 (SS) and two Group 2 (SP) neurons at P3$_L$. The boundaries of inside (barrel-center side) and outside (opposite side) were determined as described in Supplementary Fig. 1d and Methods. **g** The orientation bias index (OBI) of barrel-edge SS (eSS) neurons was significantly larger than that of barrel-edge SP (eSP) neurons at P3$_L$ ($p = 0.001$, $g = 2.135$, Welch's $t$-test: $n = 15$ eSS neurons from 7 mice and 7 eSP neurons from 4 mice). **h** Even before the initiation of AD retraction, eSS neurons already exhibited a larger OBI than eSP neurons ($p = 0.007$, $g = 1.710$, Welch's $t$-test: $n = 8$ eSS neurons with intact AD from 6 mice and 7 eSP neurons from 4 mice). **i** BD traces at P6$_L$ of the same neurons as Fig. 2f. Green shades represent individual barrels (TCA clusters). **j** The OBI of eSS neurons was significantly larger than that of eSP neurons at P6$_L$ ($p = 0.012$, $g = 2.604$, Welch's $t$-test: $n = 13$ eSS neurons from 5 mice and 4 eSP neurons from 2 mice). Box plot interpretation is described in the Methods. Scale bars: 50 μm (**a, c, d**) and 25 μm (**f**)

toward the barrel-center side as early as P3, when TCA termini start to form clusters in the barrel center (see Fig. 3d of Mizuno et al.[30]). We then repeated OBI quantification using only eSS neurons that had an intact AD at $P3_L$ (see Methods for details). Again, the OBI of these neurons was significantly larger than that of eSP neurons at $P3_L$ (Fig. 2h), although AD lengths were similar between these two groups of neurons at this age (eSS neurons with intact AD, $165.0 \pm 7.6$ μm; eSP, $166.0 \pm 12.2$ μm;

mean ± SEM). Thus, SS neurons have morphological features distinct from SP neurons, even before initiation of AD retraction. In other words, initiation of AD retraction is not the cause or trigger of SS neuron differentiation.

We then examined which aspects of BD morphology contribute to the initial orientation bias of prospective eSS neurons at P3 through detailed analysis of BD morphologies (Fig. 3). In the current study, our analyses were focused mostly on

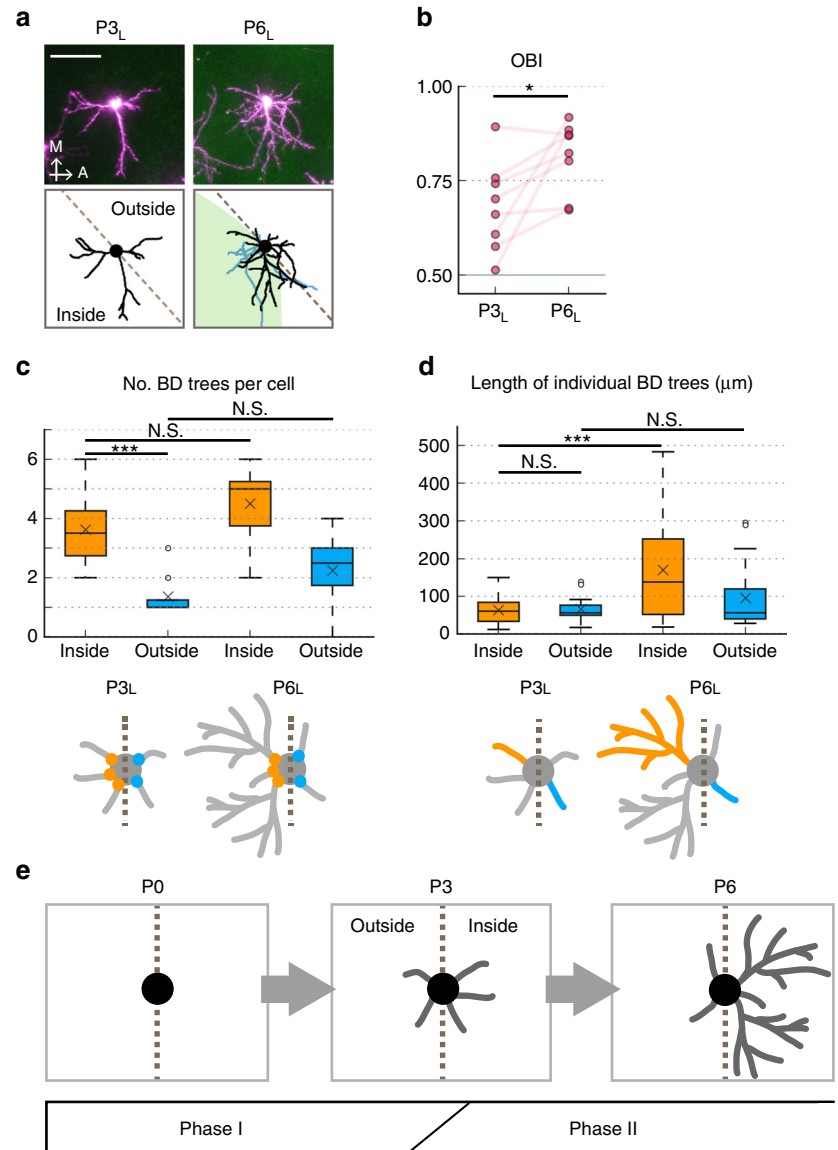

**Fig. 3** eSS neurons increased BD orientation bias by inner BD tree-specific elaboration. **a** (Top) Z-stack images of the same eSS neuron at $P3_L$ and $P6_L$. (Bottom) BDs are traced. Green shade represents the TCA cluster. Dashed lines represent the border of inside/outside. Black, BDs; Blue, axons. Scale bar: 50 μm. **b** eSS neurons increased OBI of BD between $P3_L$ and $P6_L$ ($p = 0.028$, $g = 1.244$, $n = 8$ eSS neurons of 4 mice, Paired $t$-test). **c** (Top) At $P3_L$, the number of inner BD trees was significantly larger than that of outer ones ($p < 0.001$, $r = 1.974$, Brunner–Munzel test). From $P3_L$ to $P6_L$, there were no significant increases in BD tree number both inside ($p = 0.160$, $r = 0.317$) and outside ($p = 0.137$, $r = 0.404$, Wilcoxon signed-rank test). $n = 8$ eSS neurons of four mice. (Bottom) Schematics showing origins of inner (orange) and outer (blue) BD trees. Dashed line: inside/outside border. **d** (Top) The length of individual BD trees was similar between inside and outside at $P3_L$ ($p = 0.809$, $g = 0.085$, Welch's $t$-test). The length of individual inner trees at $P6_L$ was significantly larger than that at $P3_L$ ($p < 0.001$, $g = 1.019$, Welch's $t$-test). On the other hand, there was no significant difference between the length of the individual outer trees at $P3_L$ and that at $P6_L$ ($p = 0.451$, $g = 0.401$, Welch's $t$-test). (Bottom) Schematics showing an inner tree (orange) and an outer tree (blue). Inner and outer trees are those whose origins are located inside and outside, respectively. Sample sizes are shown in Methods. $p$-values of **c** and **d** were corrected by Holm's correction. Box plot interpretation is described in Methods. **e** eSS neurons establish BD orientation bias in neonatal stages through at least two distinct phases. During Phase I (approximately between P0 and P3), eSS neurons acquire the initial orientation bias, which is ascribed to the difference of BD tree number between inside and outside. In Phase II, starting approximately at P3, the number of BD trees is not much changed but BD orientation bias is reinforced by the differential elaboration of individual inner and outer BD trees

"BD trees" rather than BD branches, because it appeared more informative for the reason described later. We categorized BD trees into two groups: "inner trees" with origins located in the barrel center-side half (inside) and "outer trees" with origins on the opposite side (outside). At $P3_L$, the number of inner trees of eSS neurons was significantly larger than that of outer trees (Fig. 3c), while lengths and tip numbers of individual trees were similar inside and outside (Fig. 3d and Supplementary Fig. 3b). These results suggest that BD orientation bias of eSS neurons at $P3_L$ is not accomplished by differential elaboration of individual inner and outer trees, but rather by the difference in the numbers of trees inside and outside the barrel. On the other hand, in eSP neurons, numbers of BD trees per cell and lengths and tip numbers of individual BD trees were similar inside and outside at $P3_L$ (Supplementary Fig. 4).

**Two phases in eSS neuron BD refinement in neonates**. Although prospective eSS neurons already showed some BD orientation bias at $P3_L$, BD morphology was still primitive at this stage, precluding more extensive directionality. However, BDs elaborated dramatically between $P3_L$ and $P6_L$ (Figs. 2e, 3a) and this was accompanied by a significant increase in OBI (Fig. 3b). We next examined which aspects of BD morphological changes influenced the OBI increment in eSS neurons between $P3_L$ and $P6_L$. To characterize large-scale dynamics, again we focused on dendritic trees rather than dendritic branches. In eSS neurons, the ratio of inner tree number to the total did not increase between P3 and P6 (Supplementary Fig. 3a). Consistently, numbers of inner and outer BD trees per cell did not increase significantly between P3 and P6 (Fig. 3c). The lengths and tip numbers of individual outer trees were also similar between $P3_L$ and $P6_L$ (Fig. 3d and Supplementary Fig. 3b). In contrast, the length and tip number of individual inner trees were significantly larger at $P6_L$ than at $P3_L$ (Fig. 3d and Supplementary Fig. 3b). These results suggest that OBI enhancement of eSS neurons between $P3_L$ and $P6_L$ primarily relies on inner BD tree-selective elaboration rather than outer BD tree-selective retraction or pruning or an increased ratio of inner to total BD tree. On the other hand, in eSP neurons, numbers of BD trees per cell and lengths and tip numbers of individual BD trees were similar inside and outside even at $P6_L$ (Supplementary Fig. 4).

Our results also suggest that eSS neurons acquire BD orientation bias in at least two phases (Phases I and II) during the first postnatal week (Fig. 3e). By P3 (Phase I), eSS neurons produced more inner BD trees than outer BD trees, although both inner and outer BD trees are morphologically simple at this stage. After P3 (Phase II), ratio of inner/outer trees from eSS neurons does not change, but inner BD trees become more elaborate.

**Differential turnover and elaboration of eSS neuron BD trees**. Next question was how eSS neuron BDs are refined during Phase II. In the initial trial, we analyzed "dendritic branch" dynamics (e.g., elongation/retraction) as we did in our previous work[30]. However, soon we noticed that this approach was not appropriate for the current work. Morphological changes of BDs in 3 days were enormous (e.g., Fig. 1f), and therefore dendritic branch analyses were too complicated to yield a meaningful outcome. Eventually, we found that focusing on "dendritic trees" rather than "dendritic branches" was more informative in understanding how the eSS neurons acquire their characteristic BD orientation bias.

We investigated the spatiotemporal dynamics of individual trees between $P3_L$ and $P5_L$ over 8-h intervals (Fig. 4). Strikingly, BD trees emerged and disappeared extensively throughout the imaging period (Fig. 4a and Supplementary Fig. 3c, d).

Emergence of new trees was not restricted to inside but observed both inside and outside (Fig. 4b). Similarly, tree elimination was not restricted to outside, but observed both outside and inside (Fig. 4c). Of outer trees newly emerged between $P3_L$ and $P5_M$ (around P5 noon), 82% (18/22) disappeared by the next imaging session (one time frame), and only a small portion of outer trees survived longer than one imaging interval (Fig. 4d, f). In contrast, only 31% (5/16) of newly emerged inner trees disappeared within one time frame, and a substantial portion survived longer (Fig. 4e, f). Thus, the survival efficiency was significantly higher for inner trees than outer trees (Fig. 4f).

We then analyzed the relationship between survival time and length of individual trees by focusing on trees newly emerged during imaging sessions. We categorized these trees into four groups: "surviving inside", "surviving outside", "eliminated inside", and "eliminated outside". Surviving trees were those present at $P5_L$ (the final imaging session), and eliminated trees were those that disappeared during imaging sessions. The lengths of surviving inner trees increased in proportion to survival time frames (Fig. 4g), while outer trees remained short. Although a few outer trees survived for a long time and elaborated, most of them extended their arbors toward the inside of the barrel (e.g., Supplementary Fig. 3e). These results suggest that long survival is preferentially conferred to some (but not all) trees that extend toward TCAs and that only these trees (mostly inner) become "winners", which subsequently elaborate over time. Importantly, winner trees were not necessarily early emerging. In other words, trees that were born later also had the chance to survive and elaborate (e.g., Fig. 4a and Supplementary Fig. 3f).

Taken together, we found that eSS neuron BD trees in the neonatal barrel cortex exhibit extensive turnover in all directions. During this dendritic tree turnover, a fraction of inner trees survived, and these surviving inner trees were extensively elaborated, suggesting that the differential dynamics of inner and outer BD trees could contribute to reinforcement of BD orientation bias between P3 and P6.

**BD tree dynamics in barrel-center SS (cSS) neurons**. Each eSS neuron receives appropriate TCA inputs using mostly inner BD trees. To understand whether this spatial bias of TCA inputs is involved in the differential dynamics of BD trees, we characterized SS neurons located in the barrel center (cSS neurons) (Fig. 5). The cSS neurons could receive TCA inputs by all BD trees, and therefore have little or no spatial bias in TCA inputs.

We found that several parameters of BD (and BD tree) growth were similar between eSS and cSS neurons, in which the data of inner and outer BD trees from eSS neurons were pooled (Supplementary Fig. 5e–h). Intriguingly, however, we found that the variance in length of individual BD trees of cSS neurons was smaller than that of eSS neurons at $P6_L$ (Supplementary Fig. 5h). To visualize the difference more clearly, we plotted the histograms (Fig. 5c, d) and cumulative curves (Supplementary Fig. 5i, j). In these analyses, inner and outer trees of eSS neurons were distinguished. In eSS neurons at $P3_L$, lengths of most inner and outer BD trees were close to the mean value (inside, 64.1 μm; outside, 67.1 μm) (Fig. 5c), while at P6, the ratio of trees with lengths close to the mean (inside, 170.1 μm; outside, 96.0 μm) was decreased both inside and outside (Fig. 5d). Instead, short trees were drastically increased both inside and outside, and long trees were increased only inside (Fig. 5d). On the other hand, most BD trees of cSS neurons had lengths close to mean values ($P3_L$, 60.8 μm; $P6_L$, 152.7 μm), and shorter and longer trees were rare at both ages (Fig. 5c, d and Supplementary Fig. 5i, j). It is important that the growth characteristics of cSS neuron BD trees differed from those of eSS neuron inner BD trees, although both could

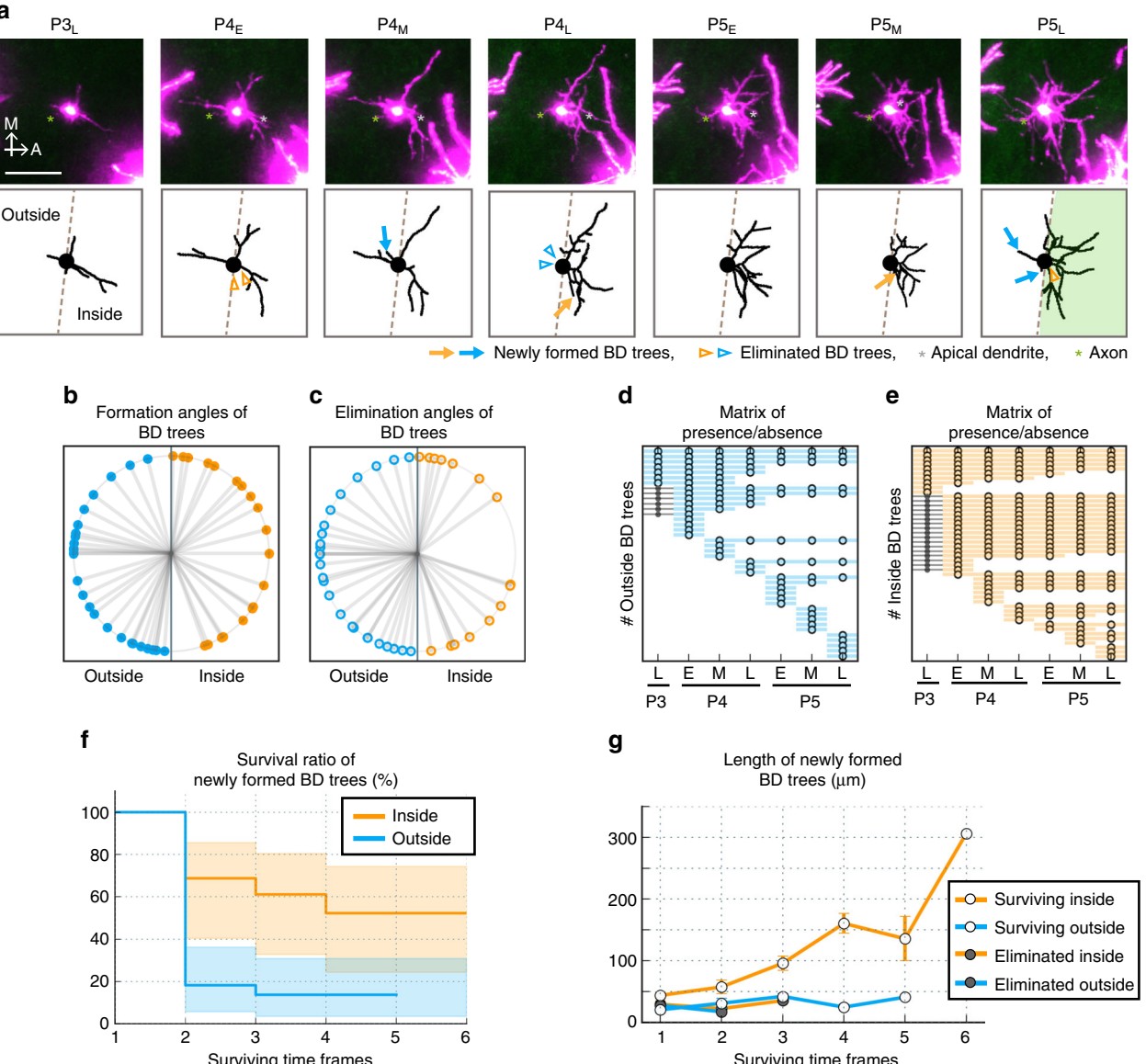

**Fig. 4** BD tree dynamics of SS neurons with 8-h intervals. **a** (Top) Z-stack images of a representative eSS neuron from $P3_L$ to $P5_L$ with 8-h interval. Gray asterisk, AD; green asterisk, axon. (Bottom) Traces of the neuron in upper panels. Arrows indicate newly formed inner (orange) and outer (blue) BD trees. Arrowheads indicate origin positions of eliminated inner (orange) and outer (blue) trees. Scale bar: 50 μm. **b, c** Angles of origins of trees that were newly formed (**b**) and eliminated (**c**) between $P3_L$ and $P5_L$. Vertical line and center indicate inside/outside border and cell body position, respectively. **d, e** Matrix of presence/absence of individual outer (**d**) and inner (**e**) trees. Each row represents an individual tree. Circles indicate time points at which the tree was present. Dots indicate time points at which the tree image was not acquired due to technical problems. Trees are sorted in order of the formation time and life span length. **f** The surviving efficiency was significantly higher for inner trees than for outer trees ($p = 0.004$, $\chi^2 = 8.282$, Log-rank test). In these analyses, only BD trees that were newly formed during $P3_L$–$P5_L$ are used. Shaded areas represent log–log transformed 95% confidence intervals. Sample sizes are shown in Supplementary Table 2. **g** Relationship between survival time frames and mean length ± SEM of individual trees that were newly emerged during imaging sessions. "Surviving In" and "Surviving Out" indicate newly emerged inner and outer trees, respectively, that remained at $P5_L$. "Eliminated In" and "Eliminated Out" indicate newly emerged inner and outer trees, respectively, that disappeared by $P5_L$. Sample sizes for **b**–**e** and **g** are shown in Methods

receive TCA inputs, suggesting the possible significance of spatial bias of TCA inputs, not just the presence of TCA inputs, in dendritic refinement dynamics.

Analyses of BD tree turnover also identified intriguing differences between eSS and cSS neurons. The BD trees of cSS neurons tended to live longer than those of eSS neurons (Fig. 5e). The numbers of eliminated trees per neuron were significantly smaller in cSS neurons than eSS neurons between $P3_L$ and $P5_L$ (Fig. 5h). cSS neurons tended to have fewer newly formed BD trees than eSS neurons, although the difference was not

significant (Fig. 5g). These results demonstrate that BD trees from cSS neurons exhibit little turnover and most live long and grow mildly. These results further suggest that dynamics of BD trees could be affected by the spatial distribution of TCA inputs.

**BD tree dynamics in pups with early infraorbital nerve (ION) cut.** ION cut is a commonly used method to interfere circuit refinement in the mouse somatosensory system (Supplementary Fig. 5a). When ION is severed in early neonates such as at P0,

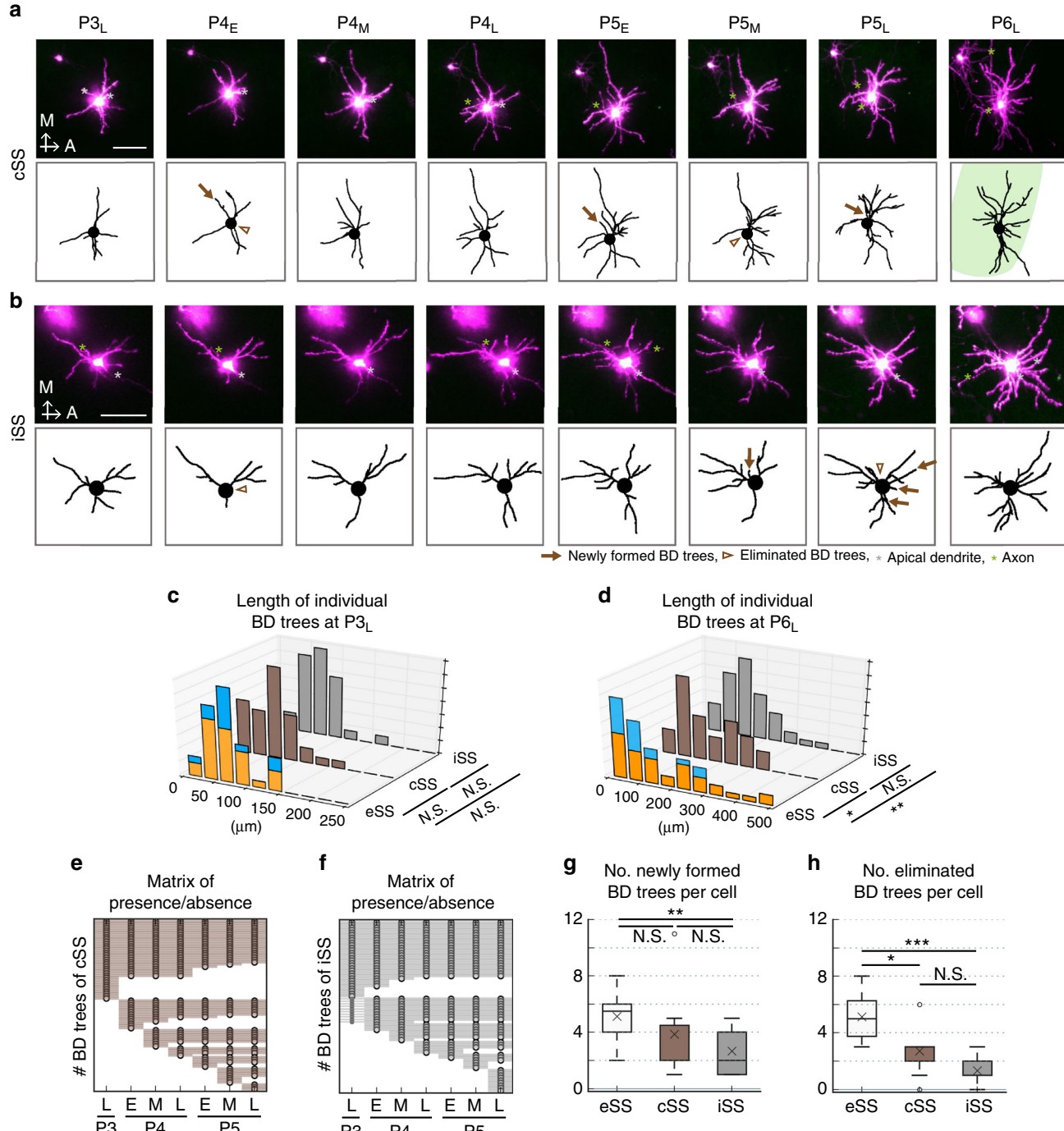

**Fig. 5** BD tree dynamics in the absence of spatial bias of TCA inputs. **a**, **b** (Top) Z-stack images of a representative barrel-center SS (cSS) neuron (**a**) and ION-cut mouse SS (iSS) neuron (**b**). Gray asterisk, AD; green asterisk, axon. (Bottom) BD morphologies of the neuron shown in the upper panel. Arrows, newly formed BD trees; arrowheads, origin positions of eliminated trees. Scale bars: 50 μm. **c**, **d** Histograms of length of individual BD trees of eSS (40 trees, 8 neurons, 4 mice; mean ± SD = 64.98 ± 36.10 μm), cSS (35, 7, 2; 60.81 ± 36.35) neurons and iSS (30, 7, 2; 62.92 ± 32.78) neurons at P3$_L$ (**c**) and eSS (54 trees, 8 neurons, 4 mice; 145.37 ± 125.26 μm), cSS (43, 7, 2; 152.68 ± 92.53) and iSS (47, 9, 3; 137.82 ± 78.24) neurons at P6$_L$ (**d**). Orange: inner BD trees. Blue: outer BD trees. At P3$_L$ (eSS vs. cSS: $p = 0.480$, $F = 1.014$; eSS vs. iSS: $p = 0.595$, $F = 1.213$; cSS vs. iSS: $p = 0.860$, $F = 1.230$, $F$-test with Holm's correction). At P6$_L$ (eSS vs. cSS: $p = 0.044$, $F = 1.832$; eSS vs. iSS: $p = 0.002$, $F = 2.563$; cSS vs. iSS: $p = 0.134$, $F = 1.399$, $F$-test with Holm's correction). See also Supplementary Fig. 5i, j. **e**, **f** Matrix of presence/absence of individual trees of 7 cSS (2 mice) (**e**) and 9 iSS neurons (3 mice) (**f**). **g** Number per cell of BD trees that were newly formed between P4$_E$ and P5$_L$ in eSS (8 neurons, 2 mice), cSS (7, 2) and iSS neurons (9, 3). eSS vs. cSS: $p = 0.310$, $r = 0.403$; eSS vs. iSS: $p = 0.008$, $r = 0.873$, cSS vs. iSS: $p = 0.464$, $r = 0.188$, Brunner–Munzel test with Holm's correction. **h** Number per cell of BD trees that were eliminated between P3$_L$ and P5$_L$ in eSS, cSS and iSS neurons. (eSS vs. cSS: $p = 0.022$, $r = 0.793$; eSS vs. iSS: $p < 0.001$, $r = 4.430$; cSS vs. iSS: $p = 0.129$, $r = 0.417$, Brunner–Munzel test with Holm's correction). Box plot interpretation is described in the Methods

barrel map formation is completely impaired[37,38] as shown in Supplementary Fig. 5b. On the other hand, the ratio of AD-possessing neurons was not different from that of normal mice at P16 (Supplementary Fig. 5c).

To understand the effects of early ION cut on BD and BD tree growth, we severed the ION at P0 and conducted TL imaging of SS neurons in these pups (Early-ION-cut mice) (Fig. 5 and Supplementary Fig. 5d). At $P3_L$, BD (and BD tree) morphology of SS neurons in ION-cut mice (iSS neurons) was similar to that of eSS and cSS neurons (Fig. 5c and Supplementary Fig. 5e–i). At $P6_L$, however, the variance in length of individual BD trees from iSS neurons was similar to that from cSS neurons but significantly smaller than that from eSS neurons (Supplementary Fig. 5h). In iSS neurons, most BD trees had lengths close to mean values (137.8 μm), while substantially shorter and longer trees were rare (Fig. 5d and Supplementary Fig. 5j). Analyses of BD tree turnover also showed that most trees of iSS neurons lived as long as those of cSS neurons (Fig. 5f). The numbers of newly formed and eliminated trees per neuron between $P3_L$ and $P5_L$ were also similar between iSS neurons and cSS neurons but significantly smaller than in eSS neurons (Fig. 5g, h). These results suggest that BD tree dynamics of iSS neurons are similar to cSS neurons.

**Early ION cut alters spatial bias of spontaneous activity.** We recently report unique features of spontaneous activity in neonatal barrel cortex L4, which shows a "patchwork"-type pattern corresponding to the barrel map. This patchwork activity is delivered to L4 via TCAs[39]. Importantly, this activity is not affected by ION cut at P4 or P5[39].

Here we examined if ION cut at P0 could affect this patchwork activity by comparing in vivo calcium signals of TCA terminals in large-barrel field of the somatosensory cortex L4 at P5 between control TCA-GCaMP6s Tg mice[39] and conspecifics receiving ION cut at P0 (Early-ION-cut mice) (Fig. 6 and Supplementary Fig. 6). In both groups, we observed spontaneous activity events, but their features were different between groups. In control mice, the boundaries of activated zones were invariable and corresponded to the boundaries of individual barrels (Fig. 6a, b, Supplementary Fig. 6b, c and Supplementary Movie 1). In contrast, they were variable in ION-cut mice (Fig. 6f, g, Supplementary Fig. 6d, e and Supplementary Movie 2). In addition, the mean area of single activated zones was more than 4.5 times larger than that of control mice (Supplementary Fig. 6f). To further examine the effects of early ION cut on spontaneous activities, we placed regions of interest (ROIs) on the large-barrel field of the somatosensory cortex of control and ION-cut mice (Fig. 6c, h). Color-coded correlation matrices constructed from the fluorescence changes (Fig. 6d, i) clearly showed that there are high-synchrony zones, which corresponded to barrels, in control mice (Fig. 6e). In contrast, such zones were not detected in the Early-ION-cut mice (Fig. 6j). The frequencies of activated events were not much different between Early-ION-cut and control mice (Supplementary Fig. 6g), although it should be noted that precise comparison of frequencies is difficult because spatial patters and sizes of activated zones were very different between two groups. These results suggest that the Early-ION-cut mouse barrel cortex still receives spontaneous activity via TCAs, but this activity no longer exhibits the patchwork pattern. Whisker-derived inputs in early neonatal stages such as P0, but not in later stages such as P4 or P5, appear necessary for patchwork patterns of spontaneous activity.

In normal mice, SS neurons located at the barrel edge (i.e., eSS neurons) receive spatially biased TCA inputs only from one direction (i.e., toward the barrel center). On the contrary, like cSS neurons in normal mice, SS neurons of ION-cut mice (i.e., iSS neurons) should not have any specific direction for proper TCA inputs. It is likely that this disruption of polarized TCA inputs results in altered dynamics of iSS neurons.

**Discussion**

To our knowledge, the present study is the first to report long-term (days) in vivo imaging analysis of dendritic dynamics in developing mammals. Herein, we have successfully accomplished in vivo imaging of the mouse barrel cortex starting at P3 and ending at P6. We revealed dynamics of SS neuron BD trees associated with circuit refinement in the neonatal barrel cortex, as well as possible involvement of spatial patterns of periphery-derived inputs in these dendritic tree dynamics as summarized in Fig. 7 (see legends). Furthermore, long-term imaging of the same neurons allowed us to retrospectively identify prospective SS neurons and characterize their features in early neonatal stages (e.g., P3) when SS neurons are indistinguishable from SP neurons by conventional methods.

ION cutting blocks the sensory inputs from whiskers[40] but not the spontaneous activity, which is most likely derived from the trigeminal ganglion or further downstream[39–42]. Instead, the patchwork-type pattern[39] of spontaneous activity is disrupted by ION-cut in early neonates (such as at P0) (Fig. 6f–j). In the normal mouse cortex, each eSS neuron receives TCA inputs (both sensory-evoked and spontaneous) from specific BD trees, which are oriented toward the barrel center. In contrast, in the cortex of ION-cut mice, SS neurons (i.e., iSS neurons) should not have any dominant BD trees to receive TCA inputs because spontaneous activity no longer exhibits a patchwork pattern. It is intriguing that BD tree dynamics of iSS neurons are similar to those of cSS neurons (Fig. 5g, h). In both cases, TCA inputs should not have specific spatial bias to SS neurons.

It appears that spatially biased TCA inputs to SS neurons facilitate cell-wide BD tree turnover, because turnover rate is higher in eSS neurons than in cSS and iSS neurons. Another scenario is that biased distributions of physical structures of TCA clusters and/or unidentified molecules derived from TCAs could induce high levels of BD dynamics in eSS neurons. Although we cannot exclude these possibilities, previous studies oppose them. For example, mice in which the gene encoding the NMDA receptor NR1 subunit is knocked out in single SS neurons showed normal TCA clusters but impaired BD refinement[30]. The *Rim1/Rim2* double knockout mice, which lack thalamocortical synaptic transmission, also show overtly normal TCA clustering but impaired BD refinement[28]. These results clearly demonstrate that the presence of physical TCA clusters are not sufficient and synaptic inputs from TCAs to SS neurons are necessary for BD refinement.

Previous in vivo imaging studies using the tectum of Xenopus tadpole or zebrafish larvae suggested that dendritic branches forming synapses with axons are stabilized and elaborated, and that glutamate receptor (NMDA and AMPA receptor)-mediated synaptic transmission is important for this stabilization process[16–18]. Our previous short-term (hours) in vivo imaging of mouse barrel cortex eSS neurons also revealed an important role for NMDA receptors in stabilization of BD branches[30]. It is likely that postsynaptic signaling induced by NMDA and/or AMPA receptors stabilizes those BD branches that make synaptic contact with appropriate TCAs, which in turn stabilizes the tree (and makes it a "winner" in the competition for thalamic inputs).

SS neurons may have homeostatic mechanisms to maintain BD tree numbers, because these numbers did not change substantially between P3 and P6 (Fig. 3c). If so, rapid elimination of unselected trees could be useful for producing more new trees

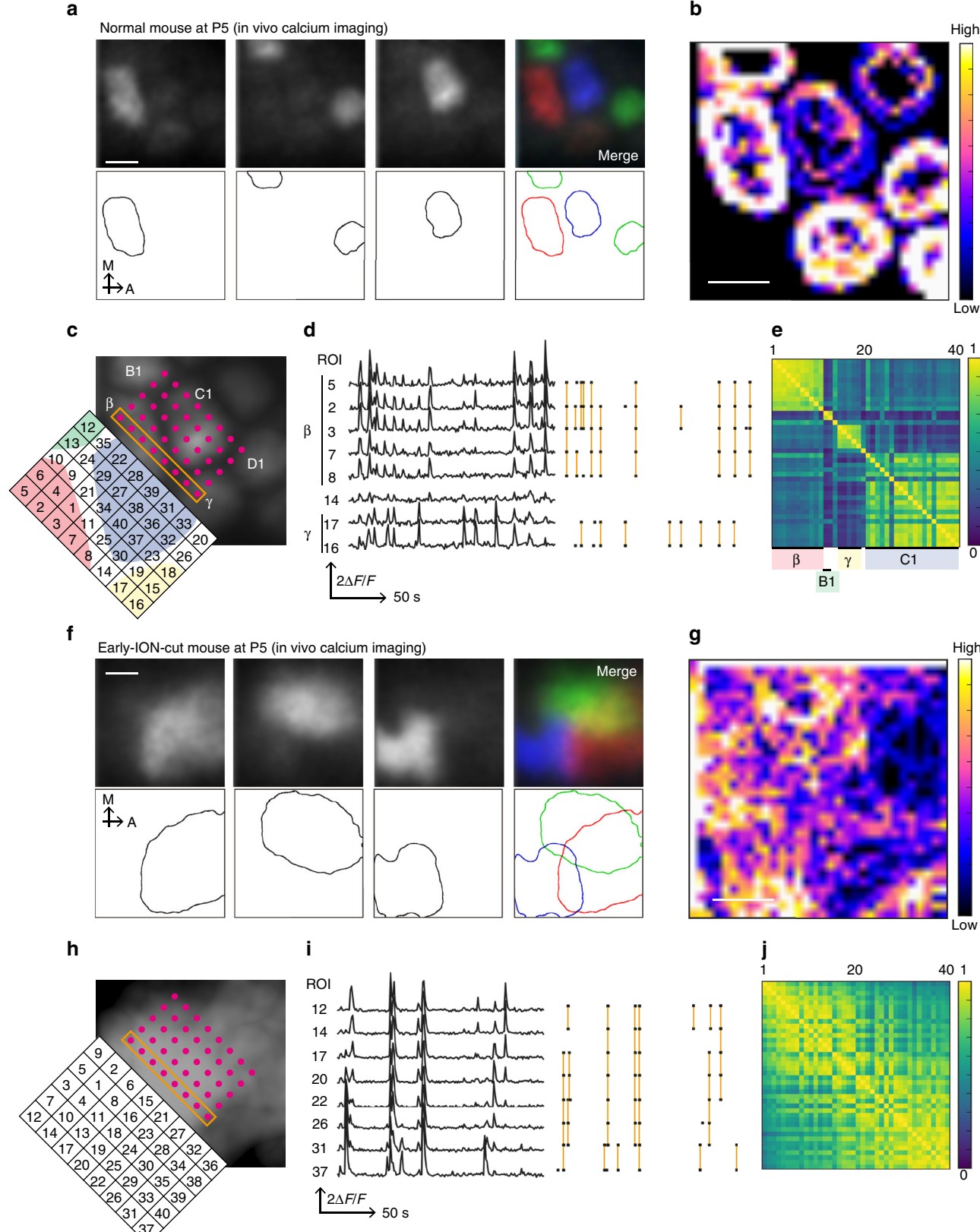

("challengers" for inputs). New winner trees continue to emerge from challenger trees predominantly near the barrel center, where appropriate TCAs are clustered. By this mechanism, the highly asymmetric pattern of eSS neuron BD projections toward the barrel center is established.

The results in the current study suggest that there are at least two distinct phases in the formation of BD orientation bias in neonates (Fig. 3e). Phase I is approximately between P0 and P3. Around P0, L4 neurons radially migrate to their final positions and start to elaborate BDs[31]. Then by P3, L4 eSS neurons create

**Fig. 6** Early ION cutting disrupts patterns of spontaneous activity. **a** In vivo calcium imaging of the large-barrel field of the somatosensory cortex L4 of a normal TCA-GCaMP Tg mouse at P5, which expresses GCaMP6s in TCAs. TCA termini demonstrated "patchwork"-type spontaneous activity[39], in which each "activated zone" corresponds to a single barrel. Representative images of activated zones at three time points and the merged image with pseudo color (top) and traces of boundaries for activated zones (bottom). See also Supplementary Movie 1. **b** A heat map of activated-zone boundaries in all activated events in 15 min. **c** Positions of circular regions of interests (ROIs). In most cases, ROIs were numbered according to the principal component analysis (PCA) scores of the correlations of the fluorescence changes. In a few ROIs (9–18), the original orders based on the PCA scores were manually rearranged to show individual barrels clearly (**e**). Color shades represent the positions of individual activated zones (i.e., barrels). **d** (Left) Representative fluorescence signals. (Right) Orange vertical lines in the raster plots ($\Delta F/F > 50\%$) demonstrate synchronized activities among ROIs, which were confined to individual barrels. **e** Correlation matrix constructed from the fluorescence changes of all ROI pairs. **f** In vivo calcium imaging of the large-barrel field of the somatosensory cortex L4 of a TCA-GCaMP Tg mouse at P5 in which ION was cut at P0 (Early-ION-cut mouse). TCA termini demonstrated spontaneous activity that lacked the patchwork pattern. Individual activated zones are largely overlapped. Representative images of activated zones at three time points and the merged image with pseudo color (top) and traces of boundaries for activated zones (bottom) are shown (see also Supplementary Movie 2). **g** Heat maps of activated-zone boundaries in all firing events observed in 15 min. **h** Positions of ROIs in the Early-ION-cut mouse. All ROIs were numbered according to the PCA scores of the correlations of the fluorescence changes. **i** Representative fluorescence signals and raster plots ($\Delta F/F > 50\%$). Orange vertical lines: synchronized activities among ROIs. Note that there were no specific clusters of ROIs. **j** Correlation matrices constructed from the fluorescence changes of all ROI pairs in the Early-ION-cut mouse. Scale bars: 150 μm

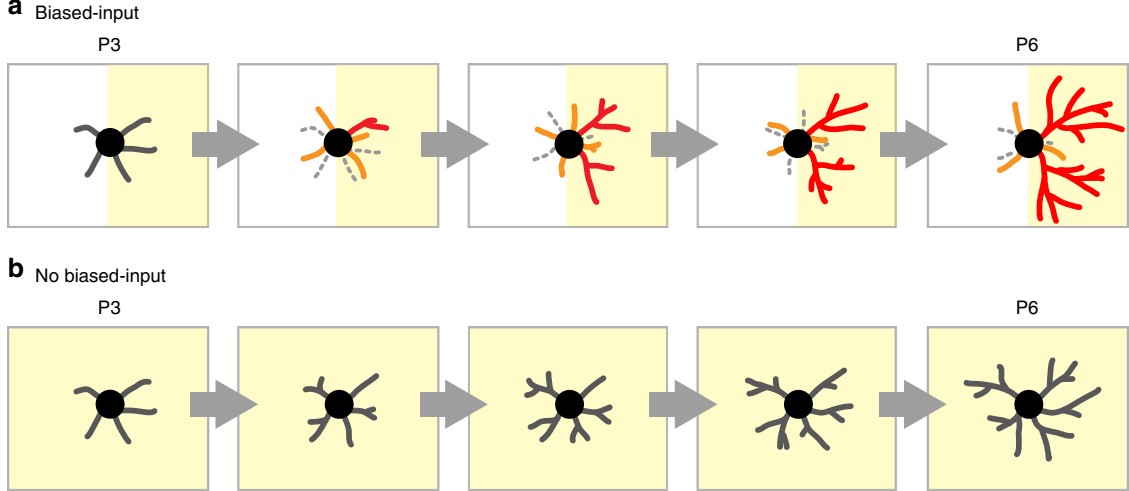

**Fig. 7** Differential BD tree dynamics in neonatal barrel cortex SS neurons. **a** In the normal neonatal mouse barrel cortex, eSS neurons receive spatially biased TCA inputs, predominantly from the barrel-center side (yellow); while on the other side (white), they receive no inputs or inputs from inappropriate TCAs. At P3, eSS neurons already have BD orientation bias, albeit weak, toward the barrel-center side (Fig. 2g). However, at this age, individual BD trees are still primitive (Figs. 2f, 3d and Supplementary Fig. 3b) and the BD orientation bias is ascribed primarily to the larger number of BD trees on the barrel side vs. the other side (Fig. 3c). BD orientation bias increases drastically between P3 and P6 (Fig. 3b), while the ratio of inner tree number to total tree number does not change (Supplementary Fig. 3a). BD trees are highly dynamic and emerge (orange) and disappear (dashed gray BDs) frequently both inside and outside the barrel center-side half (Fig. 4a–e). Meanwhile, only a fraction of trees (mostly inner) are stabilized (Fig. 4d–f) and these become extensively elaborated over time to become "winners" (red) (Fig. 4g). BD trees born later also have a chance to become winners (Supplementary Fig. 3f). **b** cSS and iSS neurons receive TCA inputs (yellow) from all directions (no biased inputs). In this situation, BD tree turnover is suppressed (Fig. 5g, h), few winners and losers are found, and most trees elaborate moderately (Fig. 5d and Supplementary Fig. 5j)

nascent orientation bias by forming more BD trees inside than outside. At this stage, both inner and outer BDs are similarly primitive. It is possible that TCAs secrete molecules that produce more BD trees on the side of the barrel center, where TCA termini form a cluster by P3, although neural activity transmitted through TCAs could also be involved. In contrast, Phase II starts approximately at P3 and ends at P6 or later. In this phase, the ratio of inner to total BD trees does not change (Supplementary Fig. 3a). Instead, stability and elaboration are primarily conferred upon a fraction of inner BD trees (Fig. 4f, g), which results in reinforcement of orientation bias toward the barrel side. Herein, we analyzed dynamic mechanisms of Phase II in detail. An important future perspective is to characterize aspects of BD refinement in Phase I, which was newly found in the current study.

Gene knockout and knockdown approaches have identified many molecules involved in SS neuron BD refinement, including the NMDA receptor, metabotropic glutamate receptor 5, adenylyl cyclase 1, protein kinase A, BTBD3, and Lhx2[19–22,30,43–46]. In future, it will be important to examine how each of these molecules is involved in BD refinement dynamics of barrel cortex SS neurons. Our Supernova system, which enables single-cell labeling and labeled cell-specific gene manipulation, could facilitate further understanding of molecular mechanisms operating in individual SS neurons[29]. It is also important to understand developmental changes of spontaneous activity in individual neurons and populations, using long-term calcium imaging (and simultaneous imaging of dendritic morphology). In the mammalian brain, developmental dendritic refinement is found not only in barrel cortex SS neurons, but also in various types of neurons in other brain areas, such as SS neurons in L4 of the visual cortex (in some animals)[2,47], Purkinje cells in the cerebellum[15,48], and mitral cells in the olfactory bulb[49]. Long-term in vivo imaging will be a powerful method for uncovering

dynamic mechanisms of dendritic refinement in these neurons and the cellular and molecular mechanisms regulating these dynamics.

## Methods

**Animals**. All experiments were performed according to the guidelines for animal experimentation of the National Institute of Genetics (NIG) and were approved by the animal experimentation committee of the NIG. To obtain pups, ICR female mice were mated with male TCA-GFP Tg mice[30] which were backcrossed from B6 to ICR more than four times or male TCA-GCaMP Tg mice[39] which were back-crossed from B6 to ICR more than one time. The day at which the vaginal plug was detected was designated as embryonic day 0 (E0) and E19 was defined as postnatal day 0 (P0). For histological analysis of ION-cut mice (Supplementary Fig. 5c), timed-pregnant ICR mice were obtained from CLEA Japan. Sex of newborn mice was not determined.

**Surgery**. In utero electroporation was conducted as described[50]. Pregnant mice at 14 postcoitus days were anesthetized with an intraperitoneal injection of sodium pentobarbital (50 mg/kg) in saline. Isoflurane was used to control anesthesia level. A midline laparotomy was performed to expose the uterus. DNA solution mixed with trypan blue (<5%, Sigma) was injected into the right lateral ventricle of embryos via a pulled glass capillary (Drummond), and square electric pulses (40 V; 50 ms) were delivered five times at the rate of one pulse per second by a CUY21SC electroporator (NepaGene). After electroporation, the uterus was repositioned, and the abdominal wall and skin were sutured. After surgery, pregnant mice were kept on a 37 °C heater until they recovered from anesthesia.

ION cutting[37,38] was performed as follows (Supplementary Fig. 5a): Pups at P0 were anesthetized with isoflurane, and a vertical incision was made on the posterior edge of the left whisker pad and the IONs were cut. After the operation, pups were kept on a warm plate and revived, after which they were returned to mothers.

Craniotomy for time lapse in vivo imaging was performed as described[30] with some critical modifications. For in vivo TL imaging of P2 and P3 mice, which are much smaller and more fragile than P5 mice, we newly designed the titanium bar that was very light (~20 mg) and small (7 × 2 × 0.5 mm) (T and I) and used the round cover glass whose diameter is 2.5 mm (Matsunami). In the morning of P2 or P3, mice were anesthetized with isoflurane. Skin covering the right hemisphere was removed using scissors to expose skull followed by applying of Vetbond (3 M) to fix the margin and to stop bleeding. Barrel area was detected by TCA-GFP signal and a small piece of bone covering the Supernova-labeled barrel area was removed with a sterilized razor blade, keeping the dura intact. Gelfoam (Pfizer) was used to stop bleeding, as necessity. To keep the brain moist, cortex buffer[51] (125 mM NaCl, 5 mM KCl, 10 mM glucose, 10 mM Hepes, 2 mM CaCl₂, and 2 mM MgSO₄; pH 7.4) was applied during surgery. After that, the window was covered with 1% low melting point agarose (Sigma) in cortex buffer and 2.5 mm diameter round cover glass. The custom-made titanium bar was glued to the skull near the window to attach mouse to two-photon microscope stage (Fig. 1b). The dental cement was applied to secure the exposed region. For analgesic and anti-inflammation, carprofen (5 mg/kg, Zoetis) was subcutaneously injected. After recovery, pups were returned to real or foster mothers.

Craniotomy for in vivo calcium imaging was performed as described[39]. P5 or P6 pups were anesthetized by isoflurane and the skull over the barrel field was removed. 1% low melting point agarose in cortex buffer[51] was applied to cover the exposed dura and the window was sealed with 3 mm diameter round cover glass (Matsunami) that secured with dental cement. A titanium bar[30] (~30 mg) was attached to the area adjacent to the cranial window. After surgery, pups were kept on a heater for recovery.

**Long-term in vivo imaging of L4 in neonatal mice**. For long-term TL imaging of L4 neurons in the large-barrel field of the primary somatosensory cortex, TCA-GFP[30] pups, in which L4 neurons were sparsely labeled by in utero electroporation-based Supernova-RFP[29] [pK036.TRE-flpe-WPRE (10–15 ng/μl) and pK037.CAG-FRT-STOP-FRT-RFP-ires-tTA-WPRE (1000 ng/μl)], were anesthetized with 0.8–1.2% isoflurane and fixed to the microscope stage using the titanium bar. Heating pad was used to keep pups warm. Images were acquired using an LSM 7MP two-photon microscope (Zeiss) with a W Plan-Apochromat 20×/1.0 DIC objective lens (Zeiss) and an LSM BiG detector (Zeiss). HighQ-2 laser (Spectra-Physics) at 1045 nm was used in most experiments. GFP and RFP were simultaneously excited and emitted fluorescence was filtered (500−550 nm for GFP and 575−620 nm for RFP). In an experiment (mouse ID #227, see Supplementary Table 1), Mai Tai eHP DeepSee titanium-sapphire laser (Spectra-Physics) running at 1000 nm was used.

Images were taken at P3_L, P4_M, P4_L, P5_E, P5_M, P5_L, and P6_L (mouse ID #231, #239, #269, #270, #356); at P2_L, P3_E, P3_M, P3_L, P4_E, P4_M, P4_L, P5_L, and P6_L (#260, #315); at P3_L, P4_L, P5_L, and P6_L (#205); at P3_L, P4_E, P4_M, P4_L, P5_L, and P6_L (#227); at P2_L, P3_E, P3_M, P3_L, P4_E, P4_M, P4_L, P5_E, P5_M, and P5_L (#313); at P2_L, P3_E, P4_L, P5_L, and P6_L (#314). PX_E, PX_M, and PX_M indicate around 4 a.m., noon, and 8 p.m. at postnatal day X (PX) as shown in Fig. 1a. Body weight was measured before or after each imaging session. Pups were returned to mothers during the interval between imaging sessions. Low body temperature, bleeding, chemical smell, and

human smell of pups could result in neglect by mothers. Therefore, pups were kept on a warm heater and with mother's bedding before returning to the mother. It was confirmed that TL-imaged pups received proper maternal care and drunk enough breast milk (Fig. 1c).

Histological analyses after the end of in vivo imaging confirmed that all analyzed neurons of both normal and ION-cut mice were located within the large-barrel field of the primary somatosensory cortex (see also Supplementary Table 1). Brain samples were prepared immediately after the P6_L imaging session. Mice were decapitated, and brains were fixed with 4% paraformaldehyde (PFA) in 0.1 M PB at 4 °C for 1–3 days. For tangential sectioning, right hemispheres were flattened and transferred to 2% PFA/30% sucrose in 0.1 M PB. Flattened cortex was kept at 4 °C for 1–2 days and tangential slices (100 μm thick) were obtained with a ROM-380 freezing microtome (YAMATO). Slices were mounted with Anti-fade Mounting Medium[52]. Images were acquired by a TCS SP5 confocal microscope (Leica). The layout of the barrel map was identified by TCA-GFP signals.

**Barrel field size measurements**. Barrel field size (Fig. 1k) was measured from confocal images by using Fiji/ImageJ 1.51p[53]. The area of the large-barrel area visualized by GFP signal of TCA-GFP mice was measured.

**Quantification of three-dimentional dendritic morphology**. Autoaligner 6.0.1 (Bitplane) was used to reduce the noise from respiratory movements. For tracing and quantification, Imaris Filament Tracer 7.0 and 8.3 (Bitplane) were used. Only neurons of which all BD and AD terminals were clearly visible were used for BD and AD analyses, respectively. Dendrite traces were generated semi-automatically and validated manually. Any dendritic processes greater than 5 μm in length was designated as a dendritic segment. Dendritic trees of which origins were in the same position between time-sequential images were considered as the same tree. AD and BDs were distinguished by their shape at the initial imaging session. Dendrites that had the same orientation as neighboring neuron ADs were judged as ADs. At early neonatal stages, neurons usually had single long thick AD toward pial surface. Axon was distinguished from BDs by the following features: (1) Axon was thinner than BD. (2) Axon emerged from the bottom of the soma. (3) Axon projected toward deep brain regions. All imaged neurons were categorized either Group 1 (SS) or Group 2 (SP) neurons as follows: Group 1 neuron was the neuron that shortened AD during imaging sessions or the neuron which had no AD at P3_L (Fig. 2b and Supplementary Fig. 2, red lines); Group 2 neuron was the neuron that continuously extended AD throughout imaging sessions (Fig. 2b and Supplementary Fig. 2, blue lines). SS neurons with intact AD (Fig. 2h) was Group 1 neurons whose ADs were still extending at P3_L.

**Quantification of BD orientation**. Because mice were detached from the two-photon microscope stage after each imaging session, orientations of acquired images were slightly different among imaging sessions. Prior to dendritic orientation analyses, these orientation artifacts among images were adjusted as follows: for each Z-stack two-photon image, neuronal coordinates were measured by Fiji/ImageJ as centroids of binary images. The central point of each image was determined by the centroid of neuronal coordinates. The central points of images were laid over and images were rotated around the central point. Angle difference between two images was calculated to minimize the error (determined by the least squares method) between coordinates of same neurons in two images. The orientation artifact was corrected by applying this angle difference to the image (e.g., white rectangles in Fig. 1d, g).

The layout of the barrel map was identified by TCA-GFP signals of in vivo images and/or tangential section images taken at P6_L (and one case at P5_L due to accidental animal death). The barrel edge was determined by the contrasting difference of TCA signal intensities in the area lying between barrel center and septa. Neurons whose cell-body center was located within 12.5 μm from the barrel edge were classified as barrel-edge neurons. Other neurons located within the barrel were classified as barrel-center neurons. The simple version of OBI, which was used in the current study, was defined as the ratio of BD segment length in the barrel-side half to the total BD length. Inside–Outside boundary, which separates barrel-side half and the other half, was determined as shown in Supplementary Fig. 1d. Each BD segment length is the length between two branch points or length between a branch point and the branch tip (or origin of the dendritic tree). An inner BD segment is the segment all or majority of whose length belongs to the barrel-side half.

**In vivo calcium imaging**. In vivo calcium imaging was performed for the large-barrel field of the primary somatosensory cortex L4 of TCA-GCaMP pups[39] at P5 or P6 under an unanesthetized condition. In these mice, L4 neurons were sparsely labeled by in utero electroporation-based Supernova-nlsRFP [pK031.TRE-Cre[30] (10 ng/μl) and pK263.CAG-loxP-STOP-loxP-nlsRFP-ires-tTA-WPRE[39] (1000 ng/μl)] as markers of the in vivo imaged areas. Images were acquired at 1 Hz (512 × 512 pixels) using an LSM 7MP two-photon microscope (Zeiss) with a W Plan-Apochromat 20×/1.0 DIC objective lens (Zeiss) and an LSM BiG detector (Zeiss). Mai Tai eHP DeepSee titanium–sapphire laser at 940 nm was used. Emitted fluorescence was filtered 500−550 nm for GCaMP6s and 575−620 nm for nlsRFP. During imaging, the body temperature of the pup was maintained using a heating

pad. When the pup head moved during imaging, these frames were excluded from the analyses.

The boundaries of the activated zones (Fig. 6a, b, f, g and Supplementary Fig. 6b −f) were determined as follows. Each image was spatially smoothed with a Gaussian filter (Sigma = 10 px). $\Delta F/F = (F-F_0)/F_0$ was calculated in each pixel at each time. $F_0$ of each pixel was obtained by averaging more than 50 images in which calcium transients were obviously absent such as under anesthesia. $\Delta F/F$ >100% were considered as activated pixels. With these criteria, boundaries of activated zones were well matched with the barrel edges in control mice. The zones that were less than 2500 $\mu m^2$ were excluded as noise.

The fluorescence intensity traces (Fig. 6d, i), which were used for raster plots (Fig. 6d, i) and correlation matrices (Fig. 6e, j), were generated from sequential 3-min images as follows. Forty ($5 \times 8$) ROIs (20 μm diameter) were positioned on large-barrel field of the somatosensory cortex L4 with 50 μm intervals (Fig. 6c, h). $F$ of ROI was obtained by averaging intensities of pixels inside the ROI. $F_0$ of each ROI was obtained by averaging more than 50 images in which calcium transients were obviously absent. For the raster plots, the threshold $\Delta F/F$ > 50% was used, because with this condition calcium transients in the control mice were most accurately detected. With the threshold $\Delta F/F$ >100%, although the barrel boundaries in control mice were most sharply visible, some apparent calcium transients (typical patchwork activity[39]) were failed to be detected (high false-negative ratio). Supplementary Movies 1 and 2 used threshold $\Delta F/F$ > 50%. Correlation matrices were sorted by principal component analysis with a few exceptions (see Fig. 6e legend). Calcium imaging data were analyzed by custom-written scripts in Python and ImageJ/Fiji.

Histological analyses after the in vivo calcium imaging confirmed that all analyzed neurons of both normal and ION-cut mice were located within the large-barrel field of the primary somatosensory cortex. Tangential sections were permeabilized and blocked in 0.2% Triton X-100/5% normal goat serum (Sigma) in 0.1 M PB. Rabbit anti-VGluT2 (1:1000; Synaptic Systems #135403) and Alexa 488-conjugated goat anti-rabbit IgG (1:1000; Invitrogen #A11034) antibodies were used. Although, in the ION-cut mouse cortex, barrel map is impaired, still identification of the large-barrel field was possible.

**Measurement of ratio of neurons with AD**. L4 neurons were labeled by in utero electroporation-based Supernova[29] [pK036.TRE-flpe-WPRE (10–15 ng/μl) and pK037.CAG-FRT-STOP-FRT-RFP-ires-tTA-WPRE (1000 ng/μl)]. P16 mice in which IONs were cut at P0 (or uncut as control) were perfused by saline and 4% PFA in 0.1 M PB before decapitated. Brains were fixed with 4% PFA in 0.1 M PB at 4 °C for 3 days. Then, brains were transferred to 30% sucrose in 0.1 M PB and kept at 4 °C for 1–2 days. Coronal slices (100 μm thick for most experiments and 50 μm thick for cytochrome oxidase (CO) staining) were obtained with a ROM-380 freezing microtome. DAPI staining (2 μg/mL; Roche) was used to determine L4 barrel field and to confirm whether IONs were cut properly. After DAPI staining, slices were mounted with Anti-fade Mounting Medium[47], and fluorescent images were acquired by a TCS SP5 confocal microscope. CO staining was also used to confirm whether IONs were cut properly. Coronal sections were incubated with CO stain solution (0.05% Cytochrome C (Sigma)/0.08% 3–3′-diaminobenzidine tetrahydrochloride (Nacalai tesque)/30% sucrose in 0.1 M PB) for 4 h at 37 °C. After visual detection of stain, sections were washed three times with 0.1 M PB and mounted with EUKITT (Kindler).

**Statistics and computing**. Two-tailed parametric and non-parametric tests were used to show the differences among means and medians, respectively (see figure legends). The asterisks in the figures indicate the following: *$p < 0.05$, **$p < 0.01$, and ***$p < 0.001$. $p < 0.05$ was considered statistically significant. $g$ and $r$ indicate the effect size for parametric and for non-parametric tests, respectively[54,55]. Error bars in bar graphs and line graphs represent SEM. In box plots, upper and lower limits of box represent 75th and 25th percentile, crosses represent mean, horizontal lines represent median, upper and lower whiskers represent maximum and minimum within 1.5 interquartile range, and observations beyond the whisker range were marked with open circles as outliers. Brunner–Munzel test was performed by R 3.2.5 and its additional package lawstat 3.0. All other analyses and visualizations were performed using Fiji/ImageJ 1.51p[53] and custom-written scripts in Python 3.5.2 with its additional packages Numpy 1.11.3, Scipy 0.18.1, Matplotlib 1.5.1, Pandas 0.19.2, Lifelines 0.9.3.2, PIL 4.2.1, OpenCV 3.3.1, Scikit-learn 0.19.1, Glob 0.6, and their later versions. Blinding was not performed. Pre-determination of sample sizes by statistical methods were not performed. At least two mice were analyzed per condition in each experiment. Exact sample sizes were provided in figure legends or following paragraphs in this section. See also Supplementary Table 1.

Figure 2b: Group 1 (SS) ($n$ = 35, 39, 38, and 33 neurons from 7, 8, 8, and 6 mice for P3, P4, P5, P6, respectively); Group 2 (SP) ($n$ = 12, 12, 12, and 8 neurons from 5, 5, 5, and 3 mice for P3, P4, P5, P6, respectively). Figure 2e: Group 1 (SS) ($n$ = 30, 38, 36, and 29 from 7, 8, 8, and 6 mice for P3, P4, P5, P6, respectively); Group 2 (SP) ($n$ = 11, 12, 11, and 7 neurons from 5, 5, 5, and 3 mice for P3, P4, P5, P6, respectively).

Figure 3d and Supplementary Fig. 3b: $n$ = 28, 12, 36, and 18 BDs for P3$_L$ inside, P3$_L$ outside, P6$_L$ inside, and P6$_L$ outside, respectively. Data were collected from eight eSS neurons of four mice.

Figure 4b–g: Data were collected from eight eSS neurons of four mice. Four neurons ($n$ = 2 mice) were analyzed every 8 h between P3$_L$ and P5$_L$, and the other four neurons ($n$ = 2 mice) were analyzed between P4$_E$ and P5$_L$ because at P3$_L$ cell morphology was not very clear due to clouding of the window. Figure 4g: Surviving inner BD trees ($n$ = 9, 9, 7, 6, 4, and 1 trees for 1, 2, 3, 4, 5, and 6 surviving time frames, respectively); Surviving outer BD trees ($n$ = 3, 3, 3, 2, and 1 trees for 1, 2, 3, 4, and 5 surviving time frames, respectively); Eliminated inner BD trees ($n$ = 7, 2, and 1 trees for 1, 2, and 3 surviving time frames, respectively); Eliminated outer BD trees ($n$ = 19 and 1 trees for 1 and 2 surviving time frames, respectively).

Supplementary Fig. 5e and f: $n$ = 15 eSS neurons from seven mice, 15 cSS neurons from five mice, and 7 iSS neurons from two mice at P3$_L$. $n$ = 13 eSS neurons from five mice, 13 cSS neurons from four mice, and 9 iSS neurons from three mice at P6$_L$. Supplementary Fig. 5g: $n$ = 8 eSS neurons from four mice, 7 cSS neurons from two mice, and 7 iSS neurons from two mice at P3$_L$. $n$ = 8 eSS neurons from four mice, 7 cSS neurons from two mice, 9 iSS neurons from three mice at P6$_L$.

Sample sizes were often different among time points for following reasons. Some neurons at some time points were excluded from the analyses because cranial windows were cloudy, and terminals of AD and/or BDs were not clearly visible. One neuron (neuron ID #231-9) was excluded from AD length analysis because its AD terminals were out of imaging range. Two neurons (neuron ID #313-1 and 2) were excluded from P6$_L$ analyses because the mouse died during the imaging session at P5$_L$ by an anesthesia problem. One neuron (neuron ID #314-1) was excluded from P3$_L$ analyses because the P3$_L$ imaging of the mouse was skipped.

**Data and code availability**. All data that support the conclusions in the study and computational code used in the study are available from the authors on reasonable request.

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

## Acknowledgements

We thank T. Hirata, T. Nakashiba, R. Iwata, and W. Luo for critical reading of the manuscript; H. Nishiyama for advice on the development of the neonatal time-lapse imaging technique; M. Miyata, T. Shimogori, A. Matsui, L. J. Lee, and R. S. Erzurumlu for advice on ION-cutting experiment; A. Kimura for valuable discussion and advice on data analysis; T. Nozaki for advice on image analysis; R. Kandasamy, S. Katori, A. Suzuki, L. Wang, and P. Banerjee for discussion; T. Sato, M. Kanbayashi, and S. Kouyama for technical assistance. This work was supported by Grant-in-Aid for JSPS Fellows JP15J03643 to S.N.; KAKENHI JP16H06143 to H.M.; KAKENHI JP16K14559, JP15H01454, JP15H04263, and Grant-in Scientific Research on Innovation Areas "Dynamic regulation of Brain Function by Scrap & Build System" (JP16H06459) from MEXT to T.I.

## Author contributions

S.N. and T.I. conceived and designed the study, and wrote the draft; S.N., H.M., and T.I. reviewed and edited the draft; H.M. designed two-photon microscope setup and conducted calcium imaging and immunohistochemistry; S.N. conducted all other experiments and all data analyses; T.I. supervised the entire project.
