## [Peer Review File · Nature Communications]

Reviewers' comments:

Reviewer #1 (Remarks to the Author):

Using the rodent whisker-barrel map formation as a model system we have gained substantial knowledge on the development of somatosensory during the past two decades. Still, little is known on how layer 4 spiny stellate neurons form their highly characteristic oriented dendritic pattern projecting towards the corresponding thalamocortical axon (TCA) cluster during the first postnatal week. Is such a pattern acquired by directed outgrowth or via lots of trial-and-error? In this manuscript, Nakazawa et al. developed the methodology necessary to conduct longitudinal imaging from postnatal day 3th to 6th to reveal the dynamics of dendritogenesis of cortical layer 4 (L4) neurons and their relationship to the whisker-related TCA clusters. This long-term imaging allowed them to reveal the following phenomena: (1) 97% of layer 4 spiny stellate neurons imaged had apical dendrites (ADs) at P3 but only 46% by P6; (2) L4 neurons with constant AD length exhibited a symmetrical dendritic pattern, while L4 neurons with reducing AD length exhibited a polarized pattern even at P3; (3) the orientation bias of L4 neurons is evident even at P3; (4) Orientation bias is established by elongation and survival of "winner" basilar dendrites (BDs); (5) Reducing thalamocortical inputs by an infraorbital nerve cut at birth decreased dendritic dynamics, including less BD turnover and reduced elongation. Based on these finding, Nakazawa et al proposed that peripheral inputs regulate the dynamics of BDs in a competitive manner to form whisker-related dendritic patterns.

It is remarkable to see such successful longitudinal imaging using very young mouse pups. Conducting nine imaging sessions at 8 hour intervals provides a wealth of dendritic dynamic data. However, as will be outlined below, the data provided in this study are insufficient to support their hypothesis that the competition from TCAs instructs the dendritogenesis of L4 SS neurons. A different sensory deprivation paradigm that allows the examination of the competitive behavior should be conducted to support this contention.

Major concerns.

1. Using the persistence or elimination of AD, authors separated L4 neurons into two groups and show clear difference in OBIs. It will be important to show whether the turnover rates of dendritic segments are different between these two groups. For example, are the numbers of winner BDs similar in both groups?

2. The authors used an infraorbital nerve (ION) cut as the sensory deprivation paradigm. Severing this nerve results in a rather dramatic reduction of TCAs and the complete absence of whisker-related patterns. The overall neurotransmission from TC synapses are likely to be greatly and uniformly reduced in such a scenario, and thus it is difficult to conceptualize competition among different connections. Thus, it is an inadequate model to make statements about the impact of competitions. A different sensory alteration paradigm that allows for examination of competition should be conducted, such as whisker cauterization for A-C rows while leaving D and E rows intact or a checker-board pattern of whisker removal.

3. "Because L4 neurons were sparsely labeled and the relative positions of neurons were roughly conserved, it was easy to identify the same neurons in images taken at different time points (imaging sessions) (Figs. 1d and 1g)." The authors also remarked on the difficulties to identify TCA clusters at P3/4 and thus the TCA patterns observed at a later time point were used to reference back for earlier time points. Such a retrospective identification is only valid if the relative locations among the L4 cells and TCAs' boundary remain the same throughout the 3 days of imaging. If L4 cells migrate toward or away from TCA clusters, then the authors cannot use P6 TCA patterns to define P3 TCA locations. It is important for the authors to examine the overall 3D relationship among all the labeled neurons during the 3 days of imaging and examine whether same degree of increases occur throughout the barrel cortex. Specifically, is there similar scaling for the barrel hollow area and the septal area?

In addition, the authors should examine whether all labeled neurons are present throughout the imaging and if there is any cell death. If yes, is there more cell loss in the septal area? This will help to answer the important question as to whether neuronal migration and/or cell death contribute to the formation of the barrel cytoarchitecture.

4. Elucidate the interactions between TCAs and dendrites at higher resolution. Is there any relationship between the proximity of TCAs to BD's and the survival rate of the latter?

Minor concerns:

1. To further understanding the dynamics of dendritogenesis, it is important to evaluate dendritic complexities by separating the different dendritic segments according to their branch orders. Particularly, the matrix of presence/absence should cluster based on branch orders.

2. Estimate the elongation/elimination speed of dendritic segments according to their branch orders.

3. Examine whether infraorbital nerve cut affect AD retractions during P3 to P6.

4. Provide more discussion explaining why ION cut reduces dendritic dynamics and speculate how dynamic dendritogenesis contributes to orientation bias.

5. For Line 300, authors stated "There was no significant difference in BD origin numbers between P3L and P6L in ION-cut mice, although they tended to increase (Fig. 5d)." Such a comparison should be conducted using paired analysis to compare the same neurons at P3L vs P6L.

6. Paragraph from Line 304 to 319 is very difficult to understand. Please edit for clarity.

7. Line 338-341, "We revealed a dynamic mechanism of dendritic refinement by which individual L4 SS neurons extend their BDs ... the role of whisker-derived inputs in BD refinement." Based on the data presented, there is not much refinement. The elimination seems to occur mainly on the dendritic segments visible for less than two imaging sessions. Perhaps it is better to emphasize that selecting a few dendritic branches located inside the barrel hollow to be the winner BDs is the major mechanism in generating the orientation bias.

8. Lines 348-351, "Importantly, though, individual BDs were still primitive at P3, and the BD orientation bias at this age was ascribed primarily by the larger number of BD origins on the barrel side (inside) versus the outside and not by differences in length and/or complexity of individual BDs between the inside and outside." At P3, the branch pattern is very primitive and thus no comparisons in complexity will be meaningful.

Reviewer #2 (Remarks to the Author):

The manuscript by Nakazawa et al. studied the remodeling of dendritic branches of layer 4 excitatory neurons in the mouse barrel cortex during the first week of postnatal development. The authors improved a chronicle imaging approach for repeated imaging of young neonate mice (P3-P6) under natural conditions (with maternal care). They showed that the surgery and early neonate imaging did not alter normal development by analyzing body weight and dendritic morphologies of L4 neurons. By two-photon time-lapse imaging over days, the authors studied changes of L4 basal dendrite morphology by analyzing orientation bias index (OBI), the number of tips and length of basal dendrite, dynamics of basal dendrites located inside/outside the barrel. Although basal dendrites grew and retracted inside and outside the barrel field, orientation bias index increased between P3 and P6, largely due to stabilization and growth of some basal

dendrites inside the barrel. Furthermore, the removal of the infraorbital nerve (ION) at P0 reduced the turnover rate of basal dendritic arbors from P3 to P6, suggesting that whisker-derived inputs increase basal dendrite dynamics.

Previous studies by the Iwasato lab (Mizuno et al., 2014) examined dendritic dynamics of layer 4 excitatory neurons in the barrel cortex of P4-P5 mice with two-photon time-lapse imaging. The current work by the same group extended previous work such that they were able to repeatedly image dendrites in P3 mice over 3 days. While improvement of the method to image younger mice is important for examining early dendritic development, the results are largely descriptive and some of the conclusions on dendritic dynamics are expected from their previous in vivo studies. The manuscript would be greatly strengthened with more in-depth mechanistic studies.

Major concerns:

(1) In my opinion, the major novel finding of the work is that ION cut animals show reduced dendritic dynamics as compared with mice with whisker inputs. This result suggests that sensory inputs to the barrel cortex are important for increasing basal dendrite dynamics. In their previous studies using NR1 cKO (Mizuno et al., 2014), the authors found increased dendritic dynamics in NR1 cKO as compared with wildtype mice, suggesting that NMDA receptor activities stabilize dendritic arbors. It is unclear why lack of sensory inputs affects dendritic dynamics differently from lack of NMDA receptor activity. One possibility is that the difference is due to the fact that ION cut affects neuronal activity globally while NR1 cKO only affects the activity of few L4 cells locally. It is also possible that ION cut may not only reduce neuronal activity but also decrease some unknown factors important for dendritic growth in the barrel cortex. The authors may want to investigate these possibilities to strengthen the significance of their work.

(2) In the abstract, the authors mentioned that " We propose that whisker-derived inputs establish the characteristic BD projection pattern of barrel cortex SS neurons by regulating competition dynamics of BDs". It is unclear to me what is the meaning or evidence for competition dynamics of BDs. The authors should be more specific about the competition. Similarly, in the discussion (the first paragraph), they wrote "We revealed a dynamic mechanism of dendritic refinement by which individual L4 SS neurons extend their BDs toward specific sets of TCAs that transmit inputs from a single whisker, as well as the role of whisker-derived inputs in BD refinement". It is also unclear what the exact mechanism is. The authors should either remove these vague terms or be more specific about them. More importantly, providing mechanistic insights into activity-dependent growth or retraction of dendritic arbors would boost the impact of the work.

Reviewer #3 (Remarks to the Author):

In the manuscript, "Competition dynamics in cortical neuron dendritic refinement revealed by long-term in vivo imaging in neonates," Nakazawa et al characterize how Layer 4 neurons in the mouse barrel cortex elaborate their dendritic arbors during early development. They use time lapse two-photon microscopy of single neurons in Layer 4 over timescales of days to image early dendritic dynamics between Postnatal day 3 to 6. They also perform deafferentation to examine how pre-synaptic axons influence these developmental dynamics.

Overall, the study is well done and very interesting and provides important insight into how Layer 4 barrel cortex spiny stellates achieve their asymmetric dendritic arbors, which will be important to begin any molecular investigations into the mechanisms. The study is largely phenomenological characterization, which is excellent, and the results are not over-interpreted. The major results are that spiny stellate cells early on already exhibit a bias in their dendritic arbors towards the center

of the barrel, that dendrites within the barrel are preferentially elaborated and stabilized, and that this requires thalamocortical axonal innervation.

The authors do well in focusing their experiments on what can be gained from 2-photon time lapse imaging and rightly acknowledge the background literature such as those using time point histological analyses, ferret studies, etc. For example, their work is excellent confirmation in another system of the existence of an initial apical dendrite in all layer 4 neurons that is eventually pruned selectively in spiny stellates.

A few critical points concern the data analyses of the orientation bias index, their descriptions of their quantifications, and their use of some quantifications to describe the phenomena:

First, their orientation bias index (OBI) as a measurement is intuitive, but is problematic when performed on early (P3) animals. Their OBI and all of their inside barrel versus outside barrel measurements is dependent on knowing the boundary of the barrel. Their images show and the authors state that the "TCA-GFP signal was too weak to clearly visualize the barrel map in vivo at this age, we used the barrel map visualized at later developmental stages (e.g., P6L) for determining the barrel boundary at P3L." Because the brain and barrels are growing (which is clear from their 2014 paper, Figure 3d), the boundary might shift by many tens of microns, which will alter many of their quantifications in the P3 animals. This probably does not grossly affect the results or interpretations of the study, but is a significant concern. Their frequent use of the schematic images (e.g., Figs 1f, 1i, 2f, 2h, etc) imply a level of precision in knowing the barrel boundary that may not exist, particularly in the P3 animal Fig 2h, with the pseudo-scale bar being 25 microns. It is not clear how the boundaries are defined despite it being a central analysis theme. Their 2014 paper is a bit more clear about how the barrel boundary is drawn, but the issue still remains about how the boundaries change over the growth of the animal. It is also not clear from their description what "barrel side" means, and if it means half of neuron that points barrel side (i.e., the dotted line in Fig S1d), or inside the barrel.

The descriptions of their quantifications can be more clear (not just how the OBI is defined) in terms of the magnitude of effects observed. For example, they state that "At P3L, the number of inner BD origins was significantly larger than that of outer ones," but it is important to state the differences for example in this case it was an average of 3.5 versus 1.5, and at P6 it is an average of 4.5 versus 2.5. With such small differences in the origin of the basal dendrites, the definition of inner versus outer barrel origin might be significant, pulling a whole branch origin into the other category. This example and throughout the paper the magnitude of the effect and the description of the quantification needs to be more standard (i.e., $n =$, size of effect, test statistic, etc).

The use of some quantifications can be better explained. For example, the Ratio of inner BD origin/total BD origin is mentioned twice in the manuscript, but only as a repeat to state that the effect was not-significant. Given that this comes right after Fig 3c, where there is a significant difference, between inner BD origin and outer BD origin, but Fig 3d is just stated in a different way (i.e., has a different rationale), the rationale for choosing this quantification should be better described. Why is this important enough not to be in the Supplementary? This data is the same as Fig 3c, but a different interpretation, why is this important enough to show? It seems like this can simply be stated in the results. Similarly, Fig 5 can also be criticized in the same way. Less than half of the Figure, and the description on Pages 18 and 19, seem to be devoted to the main message, which is the comparison of the de-afferentation and the control.

Minor points:

Language errors should be corrected, for example in the Introduction, "Major complications" are not complications at all, but advances. Etc. Most errors will probably be corrected by a copy-editor.

Page 10, line 148-149, the control P6 experiments should make clear these are static/non-time

lapse controls.

The authors mention that neurons are not analyzed if they are not “located at the barrel edge,” which creates a selection bias that the authors should briefly address, and what percentage these neurons are, etc.

Figure 2f – i need to be reversed left to right, since it is demonstrating a time course.

The schematics of the barrel boundaries in P3-P5E animals in all figures need to be more honest in their depictions of where the barrel boundary is.

Figure 4d, e, 5j, can organize the matrix better (sort on y-axis) to better depict the main message of what the authors are trying to show. At present it seems randomly organized/sorted.

Figure 5 f, g, should show the histograms in the standard way (i.e., y-axis as frequency), as is done in Fig 5 h, i. It doesn't make sense to have the same histogram and the same data in two orientations. Ideally, the normal versus ION-cut would be on the same graph/axis, as is Fig 5h and 5i.

Response to Reviewer #1

Using the rodent whisker-barrel map formation as a model system we have gained substantial knowledge on the development of somatosensory during the past two decades. Still, little is known on how layer 4 spiny stellate neurons form their highly characteristic oriented dendritic pattern projecting towards the corresponding thalamocortical axon (TCA) cluster during the first postnatal week. Is such a pattern acquired by directed outgrowth or via lots of trial-and-error? In this manuscript, Nakazawa et al. developed the methodology necessary to conduct longitudinal imaging from postnatal day 3th to 6th to reveal the dynamics of dendritogenesis of cortical layer 4 (L4) neurons and their relationship to the whisker-related TCA clusters. This long-term imaging allowed them to reveal the following phenomena: (1) 97% of layer 4 spiny stellate neurons imaged had apical dendrites (ADs) at P3 but only 46% by P6; (2) L4 neurons with constant AD length exhibited a symmetrical dendritic pattern, while L4 neurons with reducing AD length exhibited a polarized pattern even at P3; (3) the orientation bias of L4 neurons is evident even at P3; (4) Orientation bias is established by elongation and survival of “winner” basilar dendrites (BDs); (5) Reducing thalamocortical inputs by an infraorbital nerve cut at birth decreased dendritic dynamics, including less BD turnover and reduced elongation. Based on these finding, Nakazawa et al proposed that peripheral inputs regulate the dynamics of BDs in a competitive manner to form whisker-related dendritic patterns.

It is remarkable to see such successful longitudinal imaging using very young mouse pups. Conducting nine imaging sessions at 8 hour intervals provides a wealth of dendritic dynamic data. However, as will be outlined below, the data provided in this study are insufficient to support their hypothesis that the competition from TCAs instructs the dendritogenesis of L4 SS neurons. A different sensory deprivation paradigm that allows the examination of the competitive behavior should be conducted to support this contention.

(To Reviewer 1)

We appreciate the reviewer’s positive comments and useful suggestions and criticisms which

were extremely useful to improve the manuscript. We have addressed the reviewer's concerns as below.

Major concerns.

1. Using the persistence or elimination of AD, authors separated L4 neurons into two groups and show clear difference in OBIs. It will be important to show whether the turnover rates of dendritic segments are different between these two groups. For example, are the numbers of winner BDs similar in both groups?

(ANSWER)

Our results show that barrel-edge SP (eSP) neurons also exhibit BD tree turnover. The ratio of BD tree turnover of eSP neurons was not significantly different from that of eSS neurons as shown below.

eSP neurons also exhibit BD tree turnover

(a) Number per cell of BD trees that were newly formed between P4_E and P5_L in eSS (red: 8 neurons, 2 mice) and eSP (blue: 3 neurons, 1 mouse) ($p = 0.423$, $r = 0.297$, Brunner-Munzel test).

(b) Number per cell of BD trees that were eliminated between P4_E and P5_L in eSS (red: 8 neurons, 2 mice) and eSP (blue: 3 neurons, 1 mouse) ($p = 0.775$, $r = 0.092$, Brunner-Munzel test).

However, we would like not to include these data in the revised manuscript for the following reasons:

- (1) In the current study, we would like to focus on the BD refinement of SS neurons, which have biased BD projection. We believe that barrel cortex SS neuron is one of the most

useful models to understand the mechanisms underlying neural circuit refinement in the mammalian cortex. Therefore, we do not want to include the detailed analyses of SP neurons in the current manuscript.

- (2) SP neuron data described above were primitive, because sample size was small. In fact, SP neuron is the minor population (20-35%) of L4 neurons. To analyze turnover, we used only cells whose dendritic morphology is clearly visible at all imaging sessions. However, chronic imaging of neonatal cortex is still technically difficult, and it was often that cells which were clearly visible at P3 and/or P6 were not so clear at some imaging sessions. As a result, only 3 barrel-edge-SP (eSP) neurons from one mouse were available for turnover analyses.
- (3) A possible problem of Group 2 neurons is that they may contain a few SS neurons whose development is slow. If some SS neurons start to retract apical dendrite after P6, they may be categorized as Group 2 neurons, which may cause confusion.

2. The authors used an infraorbital nerve (ION) cut as the sensory deprivation paradigm. Severing this nerve results in a rather dramatic reduction of TCAs and the complete absence of whisker-related patterns. The overall neurotransmission from TC synapses are likely to be greatly and uniformly reduced in such a scenario, and thus it is difficult to conceptualize competition among different connections. Thus, it is an inadequate model to make statements about the impact of competitions. A different sensory alteration paradigm that allows for examination of competition should be conducted, such as whisker cauterization for A-C rows while leaving D and E rows intact or a checker-board pattern of whisker removal.

(ANSWER)

- (1) We recently report unique features of spontaneous activity in neonatal cortex layer 4, which shows a “patchwork”-type pattern corresponding to the barrel map (Mizuno et al., Cell Reports 2018). This patchwork activity, which is delivered to layer 4 via TCAs, was specific to the first postnatal week.

To answer the reviewer's comment, in this revision, we analyzed the effect of early ION cut on the TCA inputs. For the purpose, we used transgenic mice which express GCaMP6s in TCAs (TCA-GCaMP mice) and conducted in vivo calcium imaging of TCA terminals of these mice whose IONs were cut at P0 (**Fig. 6, Supplementary Fig. 5**). Our results clearly demonstrate that early ION cut does not block spontaneous activity. Instead, it impairs the patchwork pattern of the spontaneous activity.

We would like to insist that “competition” in this paper means competition among BD trees for TCA inputs, not competition among different TCA inputs. To manipulate this type of competition, we believe that ION-cut is an appropriate model. In normal mice, SS neurons located at the barrel edge (eSS neurons) receive TCA inputs only from one direction (i.e., from the barrel center), and therefore, “newly emerged” BD trees need to compete for TCA inputs. On the contrary, it is likely that virtually all BD trees of ION-cut mouse SS neurons (iSS neurons) have similar chance to receive TCA inputs, because the patchwork pattern was lost (see **Fig. 6**), and therefore, competition among BD trees should not be high.

Furthermore, in this revision, we analyzed SS neurons in the barrel center (cSS neurons) and found that cSS neuron BD trees showed similar dynamics (**Figs. 5e-5h**) and length distribution (**Figs. 5c and 5d**) to iSS neuron BD trees. Because cSS neurons also have no specific direction for proper TCA inputs as iSS neurons do, this result supports the importance of spatially biased TCA inputs for enhancement of BD tree dynamics.

Thus, it is likely that eSS neurons establish the characteristic BD projection pattern through biased TCA input-induced “competition” among BD trees for TCA inputs.

However, I agree with this reviewer (and Reviewer 2) that the term “competition” may be confusing. To avoid unnecessary confusion, we minimized to use “competition” in the revision. Instead, in most parts, we used the term, “selection”. In addition, to clarify these important points, we have revised the manuscript thoroughly, including Title, Abstract, Introduction,

Results, Discussion, Figures and Figure legends. The changes are marked by blue letters.

(2) We are afraid that “partial whisker cauterization or removal” may not yield a meaningful outcome. For example, if C-row whiskers are cauterized in early neonatal stage such as at P0, C-row barrels are disappeared in the cortex (e.g. Datwani, Iwasato et al., MCN 2002). If C-row whisker is cauterized in late neonatal stage such as at P5, patchwork-type spontaneous activity remains intact even in C-row barrels (our unpublished results) and dendritic orientation bias looks normal (e.g. Harris and Woolsey, J. Comp. Neurol. 196, 357-376, 1981).

Thank you so much for the insightful comment, which was extremely useful to improve the manuscript.

3. “Because L4 neurons were sparsely labeled and the relative positions of neurons were roughly conserved, it was easy to identify the same neurons in images taken at different time points (imaging sessions) (Figs. 1d and 1g).” The authors also remarked on the difficulties to identify TCA clusters at P3/4 and thus the TCA patterns observed at a later time point were used to reference back for earlier time points. Such a retrospective identification is only valid if the relative locations among the L4 cells and TCAs’ boundary remain the same throughout the 3 days of imaging. If L4 cells migrate toward or away from TCA clusters, then the authors cannot use P6 TCA patterns to define P3 TCA locations.

(ANSWER)

We agree with the review that relative position of SS neurons could shift slightly between P3 and P6 due to brain enlargement and possibly from SS neuron tangential migration. These issues hindered determination of precise barrel boundary at P3. To overcome these problems, in the current study, we used a “simple” version of orientation bias index (OBI), which divides BD segment length in the barrel-center-side half (inside) by total BD length (See **Supplementary**

Fig.1d and Methods for details).

Because our observation in the current study revealed that the relative positions of individual L4 neurons were roughly conserved between P3 and P6, we assumed that the barrel-center direction from each L4 neuron was also conserved. Approximate barrel-center direction was determined by barrel map visualized at P6.

To clarify these points, we modified passages in the Results accordingly (**p.13-14, Lines 212-222**). In addition, to avoid confusion, we drew TCA boundaries only in trace images of the final imaging session such as P6 (**Figs. 1f, 1i, 2f, 3a and 4a and Supplementary Figs. 3e and 3f**).

Thank you so much for the important comment.

It is important for the authors to examine the overall 3D relationship among all the labeled neurons during the 3 days of imaging and examine whether same degree of increases occur throughout the barrel cortex. Specifically, is there similar scaling for the barrel hollow area and the septal area?

(ANSWER)

We think that the review asks us to analyze “presumable” tangential migration of SS neurons between P3 and P6. We also eager to know that. However, according to our observation in the current study, shift of relative positions of SS neurons was subtle if any (e.g., **Fig. 1d**), which is difficult to be quantitatively analyzed with our current imaging system. For this specific purpose, we need to further improve our neonatal imaging techniques.

In addition, the authors should examine whether all labeled neurons are present throughout the imaging and if there is any cell death. If yes, is there more cell loss in the septal area? This will

help to answer the important question as to whether neuronal migration and/or cell death contribute to the formation of the barrel cytoarchitecture.

(ANSWER)

Acting upon the reviewer's suggestion, we have added new data analyzing the neuronal death. We found that all neurons observed at P3_L were present at P6_L, too, suggesting that there is no cell death during the imaging sessions. Accordingly, we have added a sentence to the Results (p.9, Lines 136-137).

4. Elucidate the interactions between TCAs and dendrites at higher resolution. Is there any relationship between the proximity of TCAs to BD's and the survival rate of the latter?

(ANSWER)

We agree with the reviewer that this is an important question. However, to analyze contact of TCAs and BDs, the resolution of our 2-photon imaging system is not sufficient. To take higher resolution images, we need to improve our imaging protocol further. For example, we need to fix pups to the 2-photon microscope stage more tightly to prevent mouse movement. Because mouse skull is so soft at P3, small motions caused by heartbeat and respiration have a big impact on high resolution imaging, which is required for visualization of TCA-BD contact. Additionally, detecting individual TCAs in vivo is difficult because all axons from the ventrobasal thalamus are labeled in TCA-GFP mouse. We intend to address this issue as a separate work in the future.

Minor concerns:

- 1. To further understanding the dynamics of dendritogenesis, it is important to evaluate dendritic complexities by separating the different dendritic segments according to their branch orders. Particularly, the matrix of presence/absence should cluster based on branch*

orders.

2. *Estimate the elongation/elimination speed of dendritic segments according to their branch orders.*

(ANSWER)

In our previous work for short-term (18 h-long) imaging (Mizuno et al., Neuron 2014), we have analyzed dynamics of “dendritic branches” (elongation/retraction). However, in the current study, the same way was not appropriate, because dendrite morphology changed enormously in 3 days and dendritic branch analyses were too complicated to yield a meaningful outcome. After a trial and error, we finally found that focusing on “**dendritic trees**” rather than “**dendritic branches**” is quite informative in understanding how the barrel edge SS (eSS) neurons acquire their characteristic BD orientation bias.

In the revised version, we clearly stated this important point (**p.17, Lines 270-277**). In addition, throughout the revised version, we clearly distinguished “dendritic trees” and “dendritic branches”.

3. *Examine whether infraorbital nerve cut affect AD retractions during P3 to P6.*

(ANSWER)

Acting upon the reviewer’s suggestion, we have added new data showing the ratio of neurons without AD. AD retraction was observed in ION-cut mice, too, and the ratio of neurons without AD was similar between normal and ION-cut mice at P16. Therefore, we conclude that ION cut had no significant effect on AD retraction.

Accordingly, we have added **Supplementary Fig. 4c** and sentences in the Results (**p.21, 349-350**) and Discussion (**p.24-25, Lines 406-409**).

4. *Provide more discussion explaining why ION cut reduces dendritic dynamics and speculate how dynamic dendritogenesis contributes to orientation bias.*

(ANSWER)

Accordingly, we have expanded the Discussion (**pp. 24-26, Lines 402-427**). Also see **p.26, Lines 428-444**.

5. For Line 300, authors stated “There was no significant difference in BD origin numbers between P3L and P6L in ION-cut mice, although they tended to increase (Fig. 5d).” Such a comparison should be conducted using paired analysis to compare the same neurons at P3L vs P6L.

(ANSWER)

This comparison was removed from the revised version, due to drastic rearrangement of the manuscript to answer many important questions raised by 3 reviewers.

6. Paragraph from Line 304 to 319 is very difficult to understand. Please edit for clarity.

(ANSWER)

We apologize for the inconvenience. We have revised (simplified) the passages accordingly (**p.19-20, Lines 318-328; p. 21, Lines 355-359**). Note that due to addition of new data for barrel-center SS (cSS) neurons, these sentences were split into two parts. The English language was edited by ENAGO (<https://www.enago.jp/>), an English language editing service of Crimson Interactive Inc..

7. Line 338-341, “We revealed a dynamic mechanism of dendritic refinement by which individual L4 SS neurons extend their BDs ... the role of whisker-derived inputs in BD refinement.” Based on the data presented, there is not much refinement. The elimination seems to occur mainly on the dendritic segments visible for less than two imaging sessions. Perhaps it is better to emphasize that selecting a few dendritic branches located inside the barrel hollow to be the winner BDs is the major mechanism in generating the orientation bias.

(ANSWER)

We have revised the manuscript accordingly (**p.24, Lines 395-398**). See also **p.26 Lines 438-**

444 and p.27, Lines 454-457.

8. Lines 348-351, “Importantly, though, individual BDs were still primitive at P3, and the BD orientation bias at this age was ascribed primarily by the larger number of BD origins on the barrel side (inside) versus the outside and not by differences in length and/or complexity of individual BDs between the inside and outside.” At P3, the branch pattern is very primitive and thus no comparisons in complexity will be meaningful.

(ANSWER)

As the reviewer suggested, we avoid using “complexity” in the revised version (**p.27, Lines 448-450**).

Response to Reviewer #2

The manuscript by Nakazawa et al. studied the remodeling of dendritic branches of layer 4 excitatory neurons in the mouse barrel cortex during the first week of postnatal development. The authors improved a chronicle imaging approach for repeated imaging of young neonate mice (P3-P6) under natural conditions (with maternal care). They showed that the surgery and early neonate imaging did not alter normal development by analyzing body weight and dendritic morphologies of L4 neurons. By two-photon time-lapse imaging over days, the authors studied changes of L4 basal dendrite morphology by analyzing orientation bias index (OBI), the number of tips and length of basal dendrite, dynamics of basal dendrites located inside/outside the barrel. Although basal dendrites grew and retracted inside and outside the barrel field, orientation bias index increased between P3 and P6, largely due to stabilization and growth of some basal dendrites inside the barrel. Furthermore, the removal of the infraorbital nerve (ION) at P0 reduced the turnover rate of basal dendritic arbors from P3 to P6, suggesting that whisker-derived inputs increase basal dendrite dynamics.

Previous studies by the Iwasato lab (Mizuno et al., 2014) examined dendritic dynamics of layer 4 excitatory neurons in the barrel cortex of P4-P5 mice with two-photon time-lapse imaging. The current work by the same group extended previous work such that they were able to repeatedly image dendrites in P3 mice over 3 days. While improvement of the method to image younger mice is important for examining early dendritic development, the results are largely descriptive and some of the conclusions on dendritic dynamics are expected from their previous in vivo studies. The manuscript would be greatly strengthened with more in-depth mechanistic studies.

Major concerns:

(1) In my opinion, the major novel finding of the work is that ION cut animals show reduced dendritic dynamics as compared with mice with whisker inputs.

(ANSWER)

Finding in ION cut experiment is important. However, it is just one of many major novel findings, which became possible by the first long-term (3 days) in vivo imaging of dendritic refinement in neonatal mammals.

Our previous short-term (18 h-long) imaging starting at P5 (Mizuno et al., Neuron 2014) provided the first *in vivo* observation of dendritic dynamics in the neonatal mammalian brain, in which small-scale dynamics (i.e., “elongation/retraction” of “dendritic branches”) were analyzed. However, this study was not sufficient, because imaging period was not long enough to cover important steps of dendritic refinement and also biased dendritic orientation was already well established at P5 (See Mizuno et al., 2014).

In our opinion, major novel findings of the current work are as follows:

(1) Long-term (3-day-long) imaging starting at P3 revealed that the barrel-edge SS (eSS) neurons establish the characteristic BD projection pattern through continuous “dendritic tree” turnover

(not just by “dendritic branch” elongation/retraction). A greater chance of longevity was given to BD trees emerged in the barrel-center side, where thalamocortical axons (TCAs) cluster.

(2) We revealed the role of spatial patterns of periphery-derived inputs in the selection dynamics in dendritic refinement.

(3) In addition, our long-term imaging of the same neurons over time allowed for “retrospective” identification of prospective SS neurons during early neonatal stages, such as P3. In these stages, most SS neurons show similar morphological features as another type of L4 excitatory neurons, star pyramid (SP). Our system can use information obtained at later developmental stages (e.g., P6) to identify prospective SS neurons at earlier stages, which enabled us for the first time to characterize features of SS neurons during early neonatal stages.

(4) Our retrospective analyses also revealed the presence of two phases in eSS neuron BD refinement during neonatal stages.

To clarify the novelty and significance of our findings in the current study, we have drastically revised the manuscript which includes Title, Abstract, Results and Discussion. Changes were marked by blue letters in the manuscript.

This result suggests that sensory inputs to the barrel cortex are important for increasing basal dendrite dynamics.

(ANSWER)

In this revised version, to further understand how TCA inputs affect the selection dynamics associated with dendritic refinement, we conducted *in vivo* calcium imaging of TCAs in the mouse in which ION was cut at P0. We found that ION cutting does not block the spontaneous activity (Fig. 6), which is most likely derived from the trigeminal ganglion or further downstream. Instead, the patchwork-type pattern of spontaneous activity was altered by ION-cut in early neonates (such as at P0) (Figs. 6c-e). Our new results in this revision suggest that

biased TCA inputs rather than just activity is critical for dendritic tree dynamics. Accordingly, we have revised manuscript.

In their previous studies using NR1 cKO (Mizuno et al., 2014), the authors found increased dendritic dynamics in NR1 cKO as compared with wildtype mice, suggesting that NMDA receptor activities stabilize dendritic arbors. It is unclear why lack of sensory inputs affects dendritic dynamics differently from lack of NMDA receptor activity. One possibility is that the difference is due to the fact that ION cut affects neuronal activity globally while NR1 cKO only affects the activity of few L4 cells locally. It is also possible that ION cut may not only reduce neuronal activity but also decrease some unknown factors important for dendritic growth in the barrel cortex. The authors may want to investigate these possibilities to strengthen the significance of their work.

(ANSWER)

As described above, ION cut does not mean that there is no activity. Instead, it disrupts patchwork-type patterns from the spontaneous activity.

We do not think that data from ION cut and those from single cell NR1KO are inconsistent. Analyses of the previous paper (Mizuno et al., 2014) and the current work are completely different. In our previous work for short-term (18 h-long) imaging (Mizuno et al., 2014), we have analyzed dynamics of “dendritic branches” (elongation/retraction). However, the same analyses were not appropriate for long-term imaging in the current work, because dendrite morphology changed enormously in 3 days. Dendritic branch analyses were too complicated to provide meaningful information. After a long trial and error, we finally found that focusing on “dendritic trees” rather than “dendritic branches” is quite informative to understand how the SS neurons acquire their characteristic orientation bias of basal dendrites. Therefore, the short-term (18-h) and long-term (3-d) imaging were analyzed differently for specific purposes and are not suitable for direct comparison.

(2) In the abstract, the authors mentioned that " We propose that whisker-derived inputs establish the characteristic BD projection pattern of barrel cortex SS neurons by regulating competition dynamics of BDs". It is unclear to me what is the meaning or evidence for competition dynamics of BDs. The authors should be more specific about the competition.

(ANSWER)

The "competition" in this paper means competition among BD trees for TCA inputs, not competition among different TCA inputs. In the current study, we report the dynamics, through which BD trees compete for TCA inputs.

However, as the reviewer suggest, the "competition dynamics" may be not a very clear word. Now we use "selection dynamics" instead. Accordingly, we have revised the manuscript including Title.

Similarly, in the discussion (the first paragraph), they wrote "We revealed a dynamic mechanism of dendritic refinement by which individual L4 SS neurons extend their BDs toward specific sets of TCAs that transmit inputs from a single whisker, as well as the role of whisker-derived inputs in BD refinement". It is also unclear what the exact mechanism is. The authors should either remove these vague terms or be more specific about them.

(ANSWER)

In this manuscript, we describe a mechanism related to dynamic aspects of dendritic refinement as explained later in this rebuttal. However, as the reviewer mentioned, the term "dynamic mechanism" may be a little vague. Therefore, we chose to use a different term, "selection dynamics" etc. Accordingly, we have revised the passages (e.g., **p.24, Line 395**).

More importantly, providing mechanistic insights into activity-dependent growth or retraction of dendritic arbors would boost the impact of the work.

(ANSWER)

To understand the mechanistic insights into activity-dependent dendritic refinement further, in the revised version, we have added two new experiments/analyses as described below:

First, to understand how the ION cut affects the activity, we conducted *in vivo* calcium imaging of the barrel cortex of ION-cut mice. We recently report “patchwork”-type spontaneous activity in neonatal barrel cortex, which is most likely derived from trigeminal ganglia or more downstream and therefore is not impaired by ION cut at late neonatal stage such as P5 (Mizuno et al., Cell Reports 2018). In the current revised manuscript, we showed that patchwork-type patterns were lost from the spontaneous activity when the ION was cut at P0 (**Fig.6** and **Supplementary Fig.5**). In this environment, each SS neuron could receive TCA inputs from virtually all dendritic trees.

Second, in the revised version, we have analyzed SS neurons located in the barrel center (cSS neurons) and compared them with SS neurons in ION-cut mice (iSS neurons) (**Fig. 5** and **Supplementary Fig. 4**). We found that cSS neurons show similar dynamics as iSS neurons. Importantly, cSS and iSS neurons have no TCA-input spatial bias to SS neurons.

Based on these results, we here propose that spatially biased TCA inputs could be important for selection dynamics of eSS neuron BD tree refinement.

If the reviewer mentioned “molecular” mechanisms, it is out of scope of this paper. However, note that precise description will be a critical basis for understanding of the molecular mechanisms in the future.

Accordingly, we have added **Fig. 6** and **Supplementary Fig.5** and expanded **Fig. 5** and **Supplementary Fig. 4**. In addition, to clarify these points, we have revised the manuscript drastically including the Abstract. Changes were marked with blue letters in the manuscript.

Thank you so much for the important comment.

Response to Reviewer #3

In the manuscript, "Competition dynamics in cortical neuron dendritic refinement revealed by long-term in vivo imaging in neonates," Nakazawa et al characterize how Layer 4 neurons in the mouse barrel cortex elaborate their dendritic arbors during early development. They use time lapse two-photon microscopy of single neurons in Layer 4 over timescales of days to image early dendritic dynamics between Postnatal day 3 to 6. They also perform deafferentation to examine how pre-synaptic axons influence these developmental dynamics.

Overall, the study is well done and very interesting and provides important insight into how Layer 4 barrel cortex spiny stellates achieve their asymmetric dendritic arbors, which will be important to begin any molecular investigations into the mechanisms. The study is largely phenomenological characterization, which is excellent, and the results are not over-interpreted. The major results are that spiny stellate cells early on already exhibit a bias in their dendritic arbors towards the center of the barrel, that dendrites within the barrel are preferentially elaborated and stabilized, and that this requires thalamocortical axonal innervation.

The authors do well in focusing their experiments on what can be gained from 2-photon time lapse imaging and rightly acknowledge the background literature such as those using time point histological analyses, ferret studies, etc. For example, their work is excellent confirmation in another system of the existence of an initial apical dendrite in all layer 4 neurons that is eventually pruned selectively in spiny stellates.

(To Reviewer 3)

We appreciate that this reviewer has supported our findings by clarifying their significance in the field. His/her insightful comments were extremely useful for us to improve the manuscripts.

A few critical points concern the data analyses of the orientation bias index, their descriptions of their quantifications, and their use of some quantifications to describe the phenomena:

First, their orientation bias index (OBI) as a measurement is intuitive, but is problematic when performed on early (P3) animals. Their OBI and all of their inside barrel versus outside barrel measurements is dependent on knowing the boundary of the barrel. Their images show and the authors state that the “TCA-GFP signal was too weak to clearly visualize the barrel map in vivo at this age, we used the barrel map visualized at later developmental stages (e.g., P6L) for determining the barrel boundary at P3L.” Because the brain and barrels are growing (which is clear from their 2014 paper, Figure 3d), the boundary might shift by many tens of microns, which will alter many of their quantifications in the P3 animals. This probably does not grossly affect the results or interpretations of the study, but is a significant concern. Their frequent use of the schematic images (e.g., Figs 1f, 1i, 2f, 2h, etc) imply a level of precision in knowing the barrel boundary that may not exist, particularly in the P3 animal Fig 2h, with the pseudo-scale bar being 25 microns. It is not clear how the boundaries are defined despite it being a central analysis theme. Their 2014 paper is a bit more clear about how the barrel boundary is drawn, but the issue still remains about how the boundaries change over the growth of the animal.

(ANSWER)

Thank you very much for the insightful comment.

The barrel edge at P6 was determined by the contrasting difference of TCA signal intensities in the area lying between barrel center and septa (**p.35, Lines 626-627**). At this age, barrel edge is so clear that it was easy to draw the barrel edge consistently with this simple method.

Regarding the changing boundary issue, please see our answer to Reviewer 1 Major concern #3.

It is also not clear from their description what “barrel side” means, and if it means half of

neuron that points barrel side (i.e., the dotted line in Fig S1d), or inside the barrel.

(ANSWER)

“Barrel side” means half of neuron that points barrel-center side (e.g. right side of the dotted line in Supplementary Fig. 1d) not “inside the barrel”. We used this simple method because the brain and barrels are growing during P3 and P6 as the reviewer mentioned above, and therefore it is impossible to determine the precise barrel boundary at early neonatal stages such as at P3. To clarify this important point, we have revised the passages accordingly (**pp.13-14, Lines 212-222**).

The descriptions of their quantifications can be more clear (not just how the OBI is defined) in terms of the magnitude of effects observed. For example, they state that “At P3L, the number of inner BD origins was significantly larger than that of outer ones,” but it is important to state the differences for example in this case it was an average of 3.5 versus 1.5, and at P6 it is an average of 4.5 versus 2.5. With such small differences in the origin of the basal dendrites, the definition of inner versus outer barrel origin might be significant, pulling a whole branch origin into the other category. This example and throughout the paper the magnitude of the effect and the description of the quantification needs to be more standard (i.e., $n =$, size of effect, test statistic, etc).

(ANSWER)

In-Out boundary, which separates barrel-side half and the other half, was determined as shown in **Supplementary Fig. 1d**. BD trees were categorized into two groups: “inner trees” with origins located in the barrel-side half (inside) and “outer trees” with origins on the opposite side (outside) (**p.15, Lines 238-241**). Note that in the revision, we use the term “BD trees” instead of “BD origins”.

To standardize the magnitude of the effect, we have shown effect size (g and r) in figure legends. Also, sample sizes (n) and p values have been shown in figure legends or methods. The test statistic can be recalculated by using these values.

The use of some quantifications can be better explained. For example, the Ratio of inner BD origin/total BD origin is mentioned twice in the manuscript, but only as a repeat to state that the effect was not-significant. Given that this comes right after Fig 3c, where there is a significant difference, between inner BD origin and outer BD origin, but Fig 3d is just stated in a different way (i.e., has a different rationale), the rationale for choosing this quantification should be better described. Why is this important enough not to be in the Supplementary? This data is the same as Fig 3c, but a different interpretation, why is this important enough to show? It seems like this can simply be stated in the results. Similarly, Fig 5 can also be criticized in the same way. Less than half of the Figure, and the description on Pages 18 and 19, seem to be devoted to the main message, which is the comparison of the de-afferentation and the control.

(ANSWER)

As the reviewer suggested, we have removed some panels from the main figures and moved to supplement figures (**Supplementary Figs. 1e-f, 3a-b, 4e-j**). Also, we shortened the description on Page 18-20: Lines 291-324 (of the original version) (**pp.21-22, Lines 346-364**).

Minor points:

Language errors should be corrected, for example in the Introduction, “Major complications” are not complications at all, but advances. Etc. Most errors will probably be corrected by a copy-editor.

(ANSWER)

Our revised manuscript was edited by ENAGO (<https://www.enago.jp/>), an English language editing service of Crimson Interactive Inc.

Page 10, line 148-149, the control P6 experiments should make clear these are static/non-time lapse controls.

(ANSWER)

As the reviewer suggested, we have rewritten the passage (**p.10, Lines 146-148**).

The authors mention that neurons are not analyzed if they are not “located at the barrel edge,” which creates a selection bias that the authors should briefly address, and what percentage these neurons are, etc.

(ANSWER)

Thank you very much for the insightful and extremely helpful suggestion to improve our manuscript.

Acting upon the review’s suggestion, we here analyzed SS neurons located at the center of the barrel (cSS neurons). As a result, we found that BD tree formation/elimination frequency was lower than that of barrel-edge SS (eSS) neurons. It was intriguing that dynamics of cSS neurons were similar to that of iSS neurons (SS neurons in the ION-cut mice). These results improve our understanding of the role of activity in the dynamic aspects of dendritic refinement in neonatal cortex. Accordingly, we have expanded **Fig. 5** and **Supplementary Fig. 4**, added sentences in the Results (**pp.19-21, Lines 311-343**) and revised the Discussion.

Figure 2f – i need to be reversed left to right, since it is demonstrating a time course.

(ANSWER)

Done.

The schematics of the barrel boundaries in P3-P5E animals in all figures need to be more honest in their depictions of where the barrel boundary is.

(ANSWER)

We have modified some figures accordingly. See above.

Figure 4d, e, 5j, can organize the matrix better (sort on y-axis) to better depict the main

message of what the authors are trying to show. At present it seems randomly organized/sorted.

(ANSWER)

As the reviewer recommended, we have rearranged the order of BD trees in matrices (**Figs. 4d, 4e and 5e**). See also **p.52, Line 980**.

Figure 5 f, g, should show the histograms in the standard way (i.e., y-axis as frequency), as is done in Fig 5 h, i. It doesn't make sense to have the same histogram and the same data in two orientations. Ideally, the normal versus ION-cut would be on the same graph/axis, as is Fig 5h and 5i.

(ANSWER)

Done.

Reviewers' comments:

Reviewer #1 (Remarks to the Author):

Comments for Nakazawa et al's revision of "Selection dynamics of cortical neuron dendrites revealed by long-term in vivo imaging in neonates"

Overall, this is a much improved version of the manuscript with new data provided and exciting findings. Nevertheless, there are some remaining concerns that should be addressed.

Major concerns:

1. Corresponding to previous major concern 1 and the replies:

I understand the technical challenges and difficulties in acquiring data for BD tree dynamics. Nevertheless, authors should be able to use their existing data to compare the length of individual BD trees between P3L and P6L for eSP neurons. Based on the information provided on Page 11 Line 167, there should be data from 12 neurons that can be summarized.

The clear separation between eSS and eSP neurons at P3 for their OBI is very novel. eSP dendritic outgrowth doesn't seem to be influenced by TCAs. If this hypothesis is correct, there should be no difference for the distribution of individual BD tree length between inside BD trees vs outside BD trees at both P3 and P6.

2. Corresponding to previous major concern 2 and the replies:

It is nice to see new calcium imaging data from TCA-GCaMP6 reporter mice included in this manuscript. In Page 22 Line 374-375, authors claimed: "In both treatment groups, we observed spontaneous activity as expected." Summary for the sizes of firing clusters was given but no functional data in terms of calcium dynamics were given. However, these data don't answer the important question if TCA activity is reduced in the S1 cortex after ION cut. The recent publication from Dr. Iwasato's lab in Cell Reports in January of 2018 showed beautiful patchwork-type spontaneous activity in the barrel cortex. In that paper, they showed original traces of $\Delta F/F$ from individual ROIs and compared the frequency changes upon treatment. It would be important to conduct similar data analysis (e.g. the frequencies of spontaneous events and the intensities of signals by showing representative calcium traces).

3. Corresponding to previous major concern 3 and the replies:

I am satisfied with using OBI instead of drawing expected TCA clusters. However, it doesn't justify defining the region with more BD trees as inside of barrels. Thus, the claim for established orientation bias toward TCAs by P3 is an over-statement unless additional data are provided to support. It is clear that SS neurons exhibit stronger OBIs than SP neurons. However, it remains to be determined whether TCAs contribute to this OBI difference.

Minor comments:

1. It may be better to use "differential dynamics" instead of "selection dynamics".

2. "When the spatial bias of TCA inputs to SS neurons was lost, BD tree turnover was suppressed and most trees became stable and elaborated mildly". This sentence may be better to be "When the spatial bias of TCA inputs to SS neurons was lost, BD trees were more stable with reduced dendritic arbors".

3. "longevity" is a term more commonly used for aging. "Stability" of dendritic tree seems to be more appropriate than "longevity".

4. For the Discussion, Page 24, Lines 405-415. It is interesting that ION cut has no impact on AD retraction while Li et al.'s study found more L4 neurons contained AD. In Li et al. studies, so called

ThVGdKO mice are double transgenic mice with a combination of total VGlut1 KO and thalamus-specific VGlut2 KO. Thus, the synaptic transmission changes will be more complex than simply decreased synaptic transmission from TCAs. Thus, I don't think it is appropriate to use the differential findings on AD retraction in the current study and that of Li et al. to conclude that "L4 neurons receive TCA inputs in ION-cut mice". I am not arguing the fact that L4 neurons can receive TCA inputs in ION-cut mice. It is the logic that I don't agree with. It seems to me that AD retractions for SS neurons may depend on the overall synaptic transmission they receive.

5. For Discussion, Page 27, Line 445-459. The orientation bias at P3 is not necessary to correlate to TCAs. Thus, I don't think there are sufficient data to claim phase I vs phase II orientation bias. There are sufficient and exciting data presented in this manuscript but they do not support the claim for different phases of orientation formation.

Reviewer #2 (Remarks to the Author):

In this revised manuscript, the authors addressed some of my previous concerns and provided new data on the changes of spontaneous calcium activity in the barrel cortex after ION cut - suggesting that this change of global calcium activity could be important for the observed dynamics of dendritic refinement. Although these modifications strengthen the work, I still feel that the work is very much descriptive. It would be much better if the authors could perturb the signaling underlying the generation of spontaneous calcium activity and link such activity to dendritic arbor dynamics.

Minor point: I suggest to change "selection dynamics" to simply "dynamics".

Reviewer #3 (Remarks to the Author):

The authors have addressed all of my concerns adequately. The calcium imaging experiments quantitation is not very well explained, but is sufficient.

Response to Reviewer #1

Comments for Nakazawa et al's revision of "Selection dynamics of cortical neuron dendrites revealed by long-term in vivo imaging in neonates"

Overall, this is a much improved version of the manuscript with new data provided and exciting findings. Nevertheless, there are some remaining concerns that should be addressed.

Major concerns:

1. Corresponding to previous major concern 1 and the replies:

I understand the technical challenges and difficulties in acquiring data for BD tree dynamics. Nevertheless, authors should be able to use their existing data to compare the length of individual BD trees between P3L and P6L for eSP neurons. Based on the information provided on Page 11 Line 167, there should be data from 12 neurons that can be summarized. The clear separation between eSS and eSP neurons at P3 for their OBI is very novel. eSP dendritic outgrowth doesn't seem to be influenced by TCAs. If this hypothesis is correct, there should be no difference for the distribution of individual BD tree length between inside BD trees vs outside BD trees at both P3 and P6.

(ANSWER)

We appreciate that the reviewer emphasized the novelty and importance of our retrospective analyses that enabled clear separation between SS and SP neurons at P3.

Acting upon the reviewer's recommendations, we have analyzed dendritic morphology details of eSP neurons at P3 and P6. Accordingly, we have added **Supplementary Fig. 4** and modified some passages (**p.14 lines 238-240, p.15 lines 257-259**). By the way, 12 neurons that the reviewer mentioned include both eSP and barrel-center SP neurons.

2. Corresponding to previous major concern 2 and the replies:

It is nice to see new calcium imaging data from TCA-GCaMP6 reporter mice included in this manuscript. In Page 22 Line 374-375, authors claimed: "In both treatment groups, we observed spontaneous activity as expected." Summary for the sizes of firing clusters was given but no functional data in terms of calcium dynamics were given. However, these data don't answer the important question if TCA activity is reduced in the S1 cortex after ION cut. The recent publication from Dr. Iwasato's lab in Cell Reports in January of 2018 showed beautiful patchwork-type spontaneous activity in the barrel cortex. In that paper, they showed original traces of $\Delta F/F$ from individual ROIs and compared the frequency changes upon treatment. It would be important to conduct similar data analysis (e.g. the frequencies of spontaneous events

and the intensities of signals by showing representative calcium traces).

(ANSWER)

As the reviewer recommended, we have added examples of original traces of $\Delta F/F$ from individual ROIs and compared the frequency changes upon treatment. Our results suggest that TCA activity is not reduced in the barrel cortex of ION-cut mice, although it is difficult to precisely compare the firing rates of normal and ION-cut mice because the spatial patterns of their firings are very different. Accordingly, we have added **Figs.6c-e, 6h-j, Supplementary Fig. 6g** and some passages in the text (**p.22, lines 375-386**) and Methods.

3. Corresponding to previous major concern 3 and the replies:

I am satisfied with using OBI instead of drawing expected TCA clusters. However, it doesn't justify defining the region with more BD trees as inside of barrels. Thus, the claim for established orientation bias toward TCAs by P3 is an over-statement unless additional data are provided to support. It is clear that SS neurons exhibit stronger OBIs than SP neurons. However, it remains to be determined whether TCAs contribute to this OBI difference.

(ANSWER)

As the reviewer mentioned, precise determination of the position of TCA clusters in the P3 cortex *in vivo* is difficult, because at this age GFP-labeled TCA clusters were not bright enough to be clearly visualized by two-photon *in vivo* imaging. However, importantly, at P3, whisker-related TCA clusters are already present in the barrel cortex layer 4, although they are not as clear as those at later stages (e.g. See Fig 3D of Mizuno, H. et al., Neuron 2014). Therefore, it appears reasonable to assume that the prospective barrel center already has higher density of TCA termini at P3. Our observation in the current study revealed that the relative positions of individual layer 4 neurons were roughly conserved between P3 and P6 (e.g. Figs. 1d and 1g). Therefore, it should be reasonable to assume that the barrel-center direction is also conserved between P3 and P6. Thus, we think that the barrel-center half (Inside), which was determined retrospectively based on the information of TCA clusters at P6, should have higher density of TCAs even at P3. To further clarify this point, we have added a passage in the Results (**p.12-13 line 210-219**) and modified **Fig. 3e**.

Minor comments:

1. It may be better to use “differential dynamics” instead of “selection dynamics”.

(ANSWER)

As this reviewer (and Reviewer 2) suggested, we now chose not to use “selection dynamics”. Accordingly, we have changed the title and text. The title is now “Differential dynamics of cortical neuron dendritic trees revealed by long-term *in vivo* imaging in neonates”.

2. “When the spatial bias of TCA inputs to SS neurons was lost, BD tree turnover was suppressed and most trees became stable and elaborated mildly”. This sentence may be better to be “When the spatial bias of TCA inputs to SS neurons was lost, BD trees were more stable with reduced dendritic arbors”.

(ANSWER)

We appreciate the reviewer’s suggestion. However, we would like to choose not to change the passage. In the current work, we found “dendritic tree turnover” in barrel neurons during neonatal stages and suggested that it may underlie the formation of characteristic BD morphology of barrel neurons. Therefore, we would like to emphasize that dendritic tree turnover is suppressed in ION-cut mice.

3. “longevity” is a term more commonly used for aging. “Stability” of dendritic tree seems to be more appropriate than “longevity”.

(ANSWER)

As the reviewer suggested, we chose not to use the term “longevity”. We now use “stability” etc.

4. For the Discussion, Page 24, Lines 405-415. It is interesting that ION cut has no impact on AD retraction while Li et al.’s study found more L4 neurons contained AD. In Li et al. studies, so called ThVGdKO mice are double transgenic mice with a combination of total VGluT1 KO and thalamus-specific VGluT2 KO. Thus, the synaptic transmission changes will be more complex than simply decreased synaptic transmission from TCAs. Thus, I don’t think it is appropriate to use the differential findings on AD retraction in the current study and that of Li et al. to conclude that “L4 neurons receive TCA inputs in ION-cut mice”. I am not arguing the fact that L4 neurons can receive TCA inputs in ION-cut mice. It is the logic that I don’t agree with. It seems to me that AD retractions for SS neurons may depend on the overall synaptic transmission they receive.

(ANSWER)

We did not intend to use the Li et al paper to conclude that “L4 neurons of ION-cut mice receive TCA inputs”. Instead, we intended to show a possible explanation for no impact of ION cut on AD retraction. However, as the reviewer suggested, Li et al paper may be a little too complicated to use for the purpose. Therefore, we have decided to delete these passages from Discussion.

5. For Discussion, Page 27, Line 445-459. The orientation bias at P3 is not necessary to correlate to TCAs. Thus, I don't think there are sufficient data to claim phase I vs phase II orientation bias. There are sufficient and exciting data presented in this manuscript but they do not support the claim for different phases of orientation formation.

(ANSWER)

Please see our answer to the Major Comment 3.

Response to Reviewer #2

In this revised manuscript, the authors addressed some of my previous concerns and provided new data on the changes of spontaneous calcium activity in the barrel cortex after ION cut - suggesting that this change of global calcium activity could be important for the observed dynamics of dendritic refinement. Although these modifications strengthen the work, I still feel that the work is very much descriptive. It would be much better if the authors could perturb the signaling underlying the generation of spontaneous calcium activity and link such activity to dendritic arbor dynamics.

(ANSWER)

Thank you so much. We agree with the reviewer that it would be nicer if we could perturb the spontaneous activity. However, before to do so, we need to uncover molecular mechanisms underlying the generation of the patchwork activity. Currently we are working hard for this goal. Hopefully, we could achieve it within a few years. Then we will be able to conduct experiments that the reviewer suggested.

Minor point: I suggest to change "selection dynamics" to simply "dynamics".

(ANSWER)

As this reviewer (and the Reviewer 1) suggested, we now chose not to use “selection dynamics”. Accordingly, we have modified text and changed the title. See our answer to the Reviewer 1 Minor comment #1.

Response to Reviewer #3

The authors have addressed all of my concerns adequately. The calcium imaging experiments quantitation is not very well explained, but is sufficient.

(ANSWER)

Thank you so much. To explain the calcium imaging experiment data better, we now added some new figures (**Figs.6c-e, 6h-j, Supplementary Fig. 6g**) and added some passages in the text (**p.22, lines 375-386**) and Methods.

REVIEWERS' COMMENTS:

Reviewer #1 (Remarks to the Author):

Authors had answered all my comments.